# The economic, agricultural, and food security repercussions of a wild pollinator collapse in Europe

Arndt Feuerbacher [1,2] ✉, Markus Kempen[3], Johannes L. M. Steidle [2,4] & Christine Wieck[2,5]

Biodiversity conservation policies often face resistance, yet the global agri-food system's vulnerability to ecosystem service declines, such as wild pollinator losses, remains poorly understood. Wild pollinators are vital for sustaining crop yields, especially nutrient-rich crops. Declines in their populations could disrupt food production, trade, and global food security. Here, we show that a hypothetical collapse of wild pollinators in Europe by 2030 would reduce European crop yields by 8%, trigger modest cropland expansion, and diminish net exports. Although global market adjustments, through changes in land use and trade, would partially mitigate these impacts, they risk exacerbating food insecurity and undermining biodiversity conservation efforts globally. Prices for pollinator-dependent crops would rise globally, with Europe seeing the steepest increases. While producers may benefit from higher prices, consumers bear the brunt. Global annual welfare losses would reach €34 billion in 2030, with Europe and the EU accounting for €24 billion and €12 billion, disproportionately impacting EU member-states resistant to biodiversity-friendly policies.

Evidence is mounting that insects and insectivorous birds are facing substantial population declines[1–12], reflecting the broader challenge of combating biodiversity loss. This decline is primarily attributed to land-use changes, intensive agricultural practices, and climate change, among other factors[7,8,10,13–15]. The conservation of wild pollinators has become a considerable political priority, especially within the European Union (EU). The EU has implemented policies and initiatives, like the EU Pollinator Initiative and the European Green Deal[16], which highlight the importance of protecting pollinators and their habitats to sustain a healthy planet supported by resilient agri-food systems and ecosystems. In Europe, all wild pollinators are insects and they provide a wide array of ecosystem services many of which are crucial for agricultural production, most notably the provision of animal-mediated pollination services[3,17–20].

The critical role of pollination services in agriculture and food security is well documented[21–25]. However, the literature so far has mainly focused on the impacts of a hypothetical global collapse of pollinators without differentiating between managed and wild pollinators[23–27]. The regional impacts of declining pollination services, particularly of wild pollinators, are not fully understood, especially regarding the adjustments through trade, consumption, and land-use changes. This study addresses this by modeling the agricultural and economic consequences of a collapse in wild pollinators that is confined to Europe. In Europe, evidence of declining insect populations is increasing[1,3,28–31], and biodiversity conservation policies have faced considerable resistance and controversy[32,33].

The crop yield changes resulting from a collapse of wild pollinators are estimated based on crop-specific dependence ratios, which

[1]Ecological-Economic Policy Modelling Research Group, University of Hohenheim, Stuttgart, Germany. [2]KomBioTa - Center of Biodiversity and Integrative Taxonomy, University of Hohenheim, Stuttgart, Germany. [3]Independent Consultant, Geilenkirchen, Germany. [4]Institute for Biology, University of Hohenheim, Stuttgart, Germany. [5]Agricultural and Food Policy Research Group, University of Hohenheim, Stuttgart, Germany. ✉e-mail: a.feuerbacher@uni-hohenheim.de

measure the extent to which yields are influenced by animal-mediated pollination. These ratios, sourced from a recent and comprehensive review[20], were combined with data on the relative contribution of wild versus managed pollinators[19] to arrive at productivity shocks that are applied across 273 European regions (Table 1). We use a detailed agricultural sector model with a particular focus on the European Union, the Common Agricultural Policy Regionalized Impact analysis (CAPRI) model, to simulate the impacts of the wild pollinator collapse on crop yields, agricultural trade, land-use, consumption and food security. This analysis provides detailed insights into regional variations in impacts across Europe and the global agri-food system. The CAPRI model is well-suited to simulate such a scenario, as it allows for adjustments on both the supply and demand sides—including shifts by producers and consumers toward crops that do not rely on pollination services. Its detailed crop-level resolution and regional disaggregation enable the model to capture heterogeneity in pollinator dependence and economic impacts. In addition, CAPRI incorporates trade linkages and price feedback mechanisms, allowing assessments of broader implications for food markets, food security, and land use beyond Europe.

Our analysis reveals an average crop yield decrease of 7.8% for the whole European continent which translates to a 0.3% decline in crop production globally. This is due to Europe's rather small weight in the global agri-food systems, comprising about 10% of the global value of pollination-dependent crops, but also because global crop land expands by 0.2%. These land expansions would potentially undermine global biodiversity conservation efforts. Furthermore, our model shows that production and prices of unaffected agricultural goods, most notably cereals and animal products, could discernibly increase. While changes in trade flows could mitigate the impact on European countries' consumption of pollination-dependent crops, such as fruits and vegetables, they risk exacerbating food insecurity and micronutrient deficiencies in low-income regions. Global welfare would decline by €34.4 billion, representing 0.5% of the total welfare generated in the global agri-food system.

## Results

The collapse of wild pollinators in Europe results in a decline of crop yields by 7.8% (with a 5.2% to 10.3% range in the confidence interval) as shown in Fig. 1. Crop production in Europe decreases by a slightly lower degree of 7.2% (4.9–9.2%) due to a slight increase in cropped area of 0.6% (0.3–1.2%). Crop production that does not depend on animal-mediated pollination services (for brevity, pollination-independent crops) is hardly impacted (0.1%), while there is a more pronounced increase in producer prices of 0.5%. In contrast, the output of pollination-dependent crop products declines by 15.5% (10.7–19.8%) and their producer prices even increase by 18.6% (12.8–24.7%). Given these strong declines, European exports of pollination-dependent crop products decline (Fig. 2), which impacts regions outside of Europe as discussed further below. Producers in regions outside of Europe, which are unaffected by the wild pollinator collapse, increase crop production, especially of pollination-dependent crops. This partially offsets the production declines in Europe and dampens the increase in prices.

In Europe, the increase in producer prices for crop products by 8.4% (5.8–11.2%) overcompensates farmers for the fall in crop yields (Fig. 1), which overall results in producer surplus gains. This result reflects the inelastic nature of agricultural supply and demand, which is embedded in the model's parameters, and accounts for substitution effects on both the supply and demand sides as well as trade responses. These mechanisms jointly explain why prices increase more than proportionally to yield losses for certain crops, resulting in income gains for some producers despite falling production (see also the discussion of the King-Davenant-Law below). The price increases also lead to an endogenous change in crop yields and incentivize land

reallocation from low-input to high-input agricultural production technologies. The combined effect of these model mechanisms buffers the simulated production shocks for pollination-dependent crops by about 2.1% on average across all European regions (Supplementary Table 1).

Oilseed, vegetable and fruit crops are most dependent on pollination services. Among oilseeds, Rapeseed and Sunflower are most impacted. While Rapeseed has a lower dependence on pollination services than Sunflower, it has a much higher dependence on wild pollinators (Table 1), resulting in substantially higher declines in yield and output (Fig. 1). Comparatively, the increase in Rapeseed prices is rather low, which is also due to strong trade linkages to the rest of the world, as shown by the trade indicators (Table 1). While Europe produces almost two-thirds of global Sunflower production, this primarily serves the European domestic market. The resulting weak trade linkages explain the disproportionately high increases in Sunflower prices relative to the decline in yield.

The strong yield reductions in the commodities of Other Vegetables, Apples, and Other Fruits (Table 1) lead to the largest absolute declines in production across Europe (Fig. 2). These commodities together also account for almost two-thirds of the production value of pollination dependent crops in Europe (Table 1). The extent to which global agri-food markets can mitigate these declines varies. For the commodity Other Fruits, strong trade linkages between Europe and non-European countries result in marked production increases outside of Europe (Fig. 2). As a result, despite the sharp drop in European production, trade nearly compensates for Europe's production shortfall, substantially softening the rise in producer prices (Fig. 1) and causing only a slight reduction in the consumption of Other Fruits. In contrast, Europe is less reliant on imports of Other Vegetables and Apples (Table 1). Therefore, the yield declines in these products prompt only a relatively low production response outside Europe, leaving most of the European production losses unaddressed, which leads to pronounced declines in consumption within Europe (Fig. 2), while producer prices increase sharply (Fig. 1).

### Aggregate production and welfare impacts on the Global agri-food system

Even though the European agricultural sector comprises only a moderate share in the global agri-food system, the collapse of wild pollinators in Europe results in non-trivial impacts on countries and trade-bloc regions in other continents (Fig. 2). The asymmetric decline in crop output in Europe leads to higher crop prices globally (Fig. 1) which incentivizes increases in crop production outside of Europe (Fig. 2). This particularly benefits farmers in Central- and South America where output increases are reported to be the highest (Fig. 3A). The increases in crop prices benefit farmers and result in higher farm profits in almost all regions (Supplementary Fig. 1) that add up to a global increase in producer surplus of €4.6 billion with a 95% confidence interval of €0.8 billion to €9.5 billion (Supplementary Table 2). These gains rise with the magnitude of the productivity shock, reflecting stronger price responses under more severe pollinator losses. Hence, globally, producers benefit from negative productivity shocks confined to Europe, as prices increase disproportionately.

Higher crop prices also hurt consumers and lead to a decline in total consumed quantity of crops, resulting in a global mean welfare loss of €34.4 billion (€23.2–€44.0 billion), which is a relative welfare decline of 0.5% (0.3–0.7%) compared to the agri-food sector's welfare contribution in the reference scenario (Supplementary Table 2). The model only captures the agri-food system and thus no other economic sectors. In contrast to the whole world economy, the reported welfare decline comprises 0.05% (0.03–0.06%) of global GDP in the base year (2017). The relative welfare changes are spatially divergent (Fig. 3B). Unsurprisingly, European regions experience the

**Table 1 | Mean productivity shocks for pollination-dependent agricultural commodities in Europe following a collapse of wild pollinators**

| Group | Commodity accounts in CAPRI | Europe's share in global production value | Share in total pollination dependent crop production value within Europe (Σ = 100%) | Trade indicators (adjusted for intra-regional trade)[b] Self-sufficiency ratio | Export intensity | Import penetration ratio | Mean dependence ratio | Relative contribution of wild pollinators %-share of dependence ratio | Productivity shock after collapse of wild pollinators in Europe Mean | Lower bound | Upper bound |
|---|---|---|---|---|---|---|---|---|---|---|---|
| | | In % | In % | In % | In % | In % | In % | | In % | In % | In % |
| Other cereals and crops | Other Cereals[a] | 13.7 | 1.6 | 76.3 | 0.3 | 23.9 | 18.7 | 23.0 | 4.3 | 2.2 | 6.5 |
| | Pulses | 9.1 | 2.2 | 103.4 | 13.1 | 10.2 | 9.2 | 76.0 | 7.0 | 4.2 | 9.1 |
| Oilseeds | Rapeseed | 32.6 | 6.0 | 87.3 | 26.6 | 35.9 | 27.0 | 59.6 | 16.1 | 4.1 | 36.1 |
| | Sunflower | 64.7 | 5.9 | 104.6 | 11.1 | 7.0 | 54.0 | 14.7 | 7.9 | 2.6 | 12.3 |
| | Soya | 5.0 | 2.0 | 50.2 | 18.0 | 58.8 | 19.0 | 50.4 | 9.6 | 2.1 | 16.6 |
| Vegetables and Fruits | Tomatoes | 24.8 | 16.0 | 99.0 | 7.8 | 8.7 | 27.0 | 50.4 | 13.6 | 3.0 | 23.3 |
| | Other Vegetables | 7.5 | 39.3 | 91.3 | 1.6 | 10.2 | 38.1 | 76.0 | 28.9 | 25.3 | 32.2 |
| | Apples, Pears and Peaches[c] | 14.5 | 5.3 | 93.7 | 8.3 | 14.1 | 66.6 | 42.9 | 28.6 | 7.2 | 37.1 |
| | Citrus Fruits | 6.1 | 2.9 | 45.4 | 2.0 | 55.5 | 51.3 | 15.5 | 8.0 | 6.1 | 9.7 |
| | Other Fruits and Nuts[c] | 8.0 | 18.8 | 57.7 | 9.1 | 47.5 | 61.0 | 48.1 | 29.4 | 23.1 | 33.3 |

The productivity shocks are calculated as percentage reductions in yield, based on mean dependence on pollination services and the relative contribution of wild pollinators. Lower and upper bounds represent a 95% confidence interval, assuming a triangular distribution. The detailed region-specific shocks are reported in Supplementary Table 3.
Source: Own calculations based on Siopa et al. (2024), Reilly et al. (2024), and production data obtained from the FAOstat database[85]. Note: Commodity disaggregation follows the CAPRI model. Pollination-dependent commodities not grown in Europe (e.g., oil palm, coffee, and cocoa) are excluded.
[a]Other Cereals include only one pollination dependent crop, buckwheat, a pseudo-cereal. For this crop no data was available to estimate the 95% confidence interval, instead a variation by ± 50% was assumed.
[b]Indicators are calculated following the OECD[87] for 2017 based on the CAPRI model database. Self-sufficiency is the ratio between domestic production quantity and domestic demand. Export intensity is calculated by dividing the export quantity by domestic production. Import penetration ratio is the share of imports in total demand.
[c]Abbreviated henceforth as "Apples" and "Other Fruits", respectively.

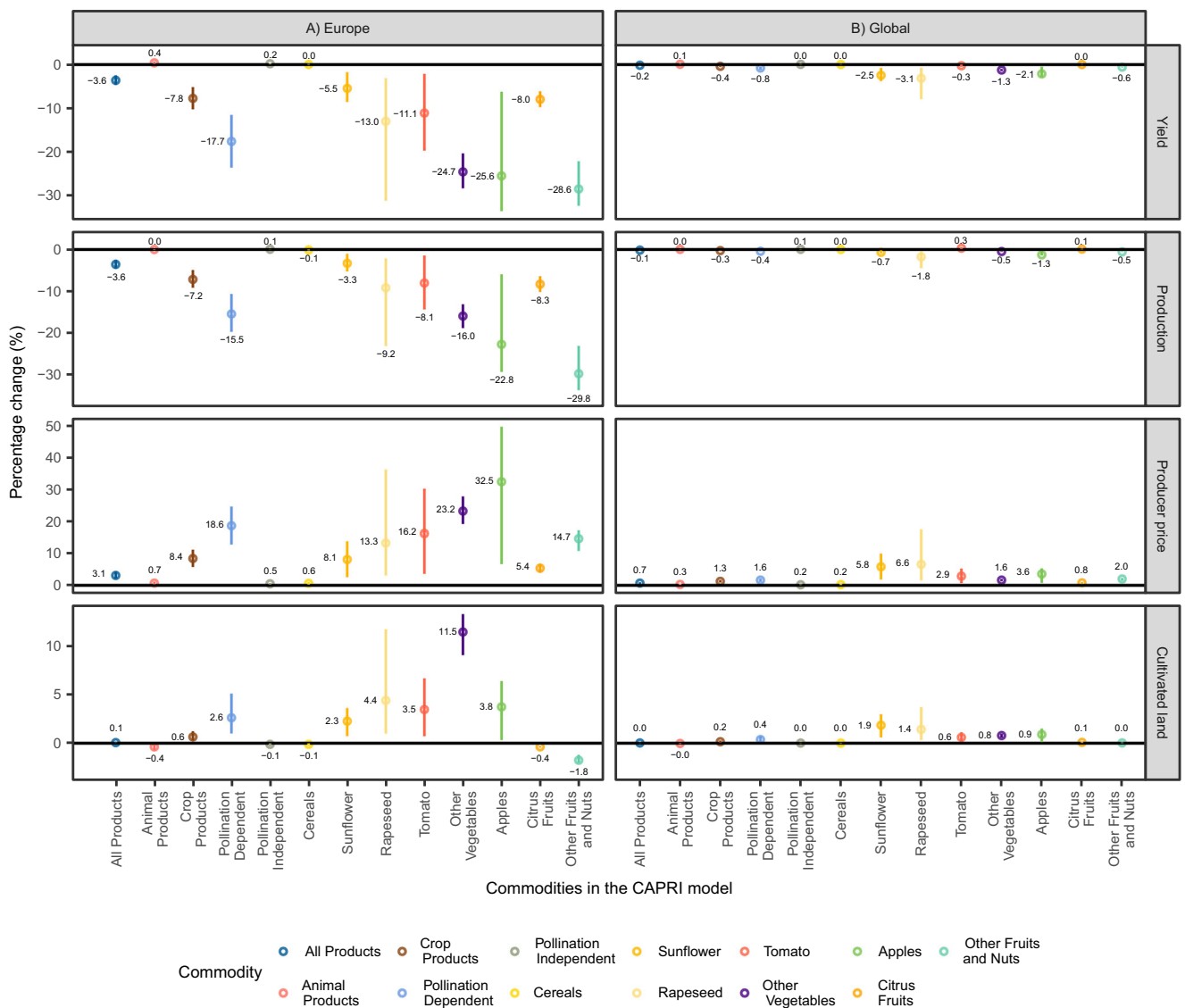

**Fig. 1 | Commodity-level impacts of a European wild pollinator collapse on yields, production, prices, and land use.** Percentage changes are reported for yield, production, producer prices, and cultivated land within (**A**) Europe and (**B**) globally, following a simulated collapse of wild pollinators in Europe. Each dot represents the CAPRI model output under the mean productivity shock scenario. The vertical lines show the range of results from three (*n* = 3) scenarios that reflect different levels of productivity shock, corresponding to the mean and the bounds of the 95% confidence interval. Results are model-derived and do not reflect standard deviation or standard error. The unit of analysis is the simulated output at the commodity level (e.g., cereals, vegetables) per region. All values are reported as percentage changes relative to a baseline simulation without pollinator collapse. The three scenarios were derived from biologically plausible variations in productivity losses due to pollinator decline and are treated as independent simulations. Note: The y-axis extent is specific to each indicator. Aggregate changes in yield, production, and producer prices were weighted by the base production values. Aggregate changes in land use are based on physical area. See Methods for scenario design and Supplementary Materials for source data.

highest loss in welfare within the agri-food system, declining by 2.2% or €23.8 billion. Countries with particularly high dependence on wild pollinators report relative welfare declines up to 8.6%, as is the case in Ukraine. Within the EU, Spain experiences the highest welfare decline of 3.7%. Regions outside of Europe observe only very moderate welfare declines or, in few instances, even slight increases. Outside of Europe, welfare within the agri-food sector declines by 0.2% or €10.6 billion (Supplementary Table 2).

**Impacts on food security**
The decline in yields of pollination-dependent crops leads to a reduction in crop production across Europe, especially for nutrient-dense crops like vegetables and fruits (Fig. 2). This drop in production is partly mitigated by international trade. Previously a net exporter, the

EU now becomes a net importer of vegetables and fruits (Supplementary Fig. 2). Non-EU European countries remain net importers, with their net imports of fruits and vegetables by 11.9%. Europe's rising demand is largely met by increased exports from Asia and Central and South America, accounting for 60% and 20% of the total increase in European net imports, respectively.

Despite these trade adjustments, Europe experiences a non-trivial decline in overall food intake, especially in micronutrient availability. Folate and particularly Vitamin A availability drop sharply, with the latter dropping across all food by 3.7% within the EU and 3.8% in non-EU Europe (Fig. 4). The upper bound of the 95% confidence interval suggests these declines could be about twice as high. The food security implications of this reduction in nutrient-rich foods are substantial. Approximately 58 million Europeans, living predominantly outside the

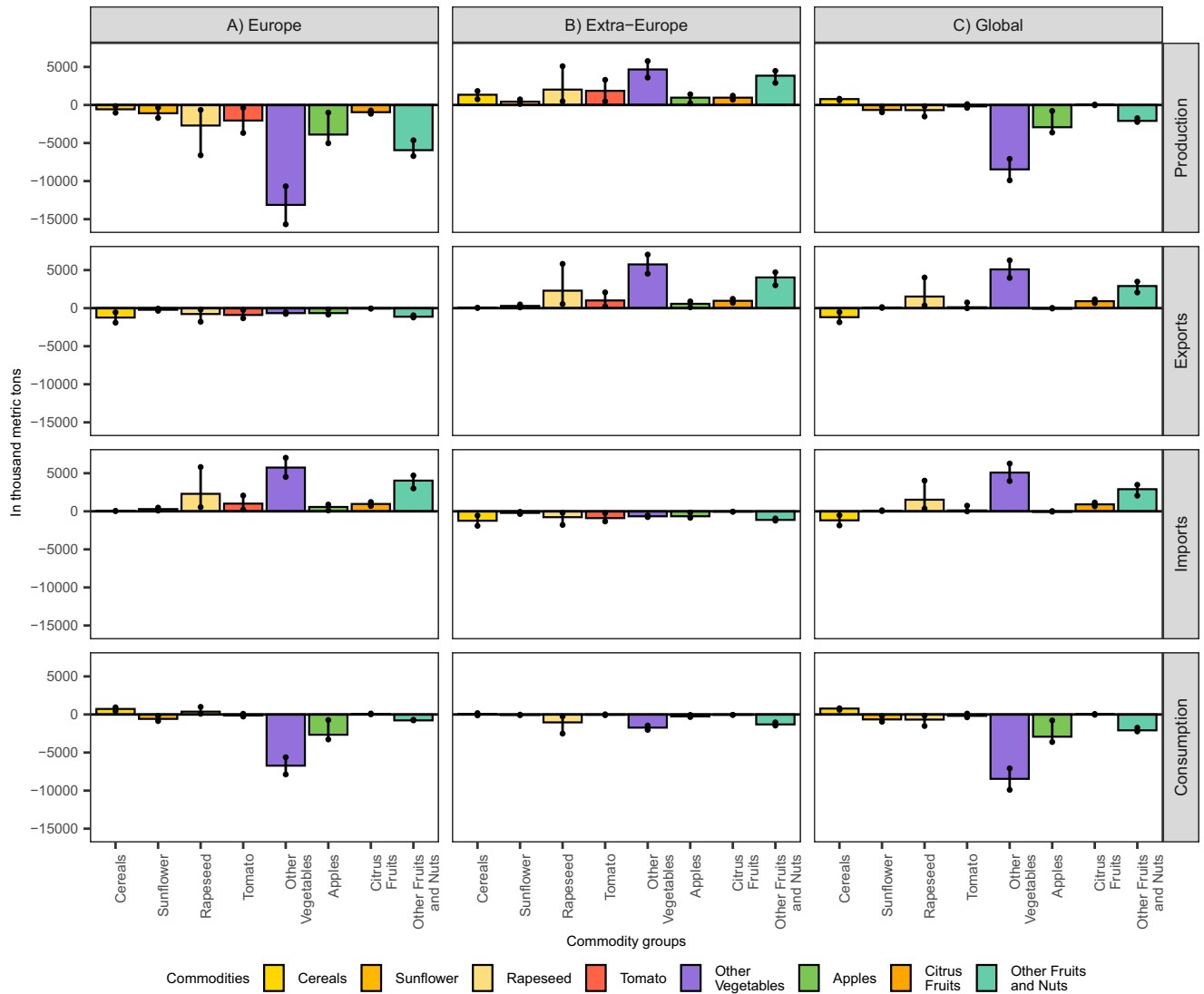

**Fig. 2 | Changes in production, exports, imports, and consumption of crops following a simulated collapse of wild pollinators in Europe.** Each bar represents the CAPRI model output for the mean productivity shock scenario. Dots indicate the range of results from two additional simulations corresponding to the upper and lower bounds of the 95% confidence interval for productivity losses in pollinator-dependent crops. In total, three independent scenario simulations were conducted ($n = 3$). Exports and imports are adjusted to exclude intra-regional trade. Results are shown in thousand metric tons and are aggregated for (**A**) Europe, (**B**) Extra-Europe, and (**C**) Global regions. Extra-Europe refers to all regions outside of Europe. All values represent differences relative to a baseline scenario without pollinator collapse. See Methods for scenario design and Supplementary Materials for source data.

European Union, face moderate or severe food insecurity, with a majority (41 million) living in Southern or Eastern Europe, where the prevalence of moderate food insecurity ranges from 10% to 30% of the population[34]. Rising prices for nutrient-dense foods, coupled with decreased intake and micronutrient availability, would likely intensify Europe's food insecurity challenges.

Globally, food insecurity is already concentrated in regions outside Europe. As Europe's import demand for vegetables and fruits rises, it intensifies competition for these foods on international markets. Although some of these regions may increase production, the surge in European demand more than offsets those gains. The result is a net decline in the availability and consumption of nutrient-dense foods worldwide (Fig. 2). This exacerbates food insecurity in regions already vulnerable to malnutrition[34], particularly in Africa, as well as parts of Central and South America and Asia (Supplementary Fig. 3).

In summary, a hypothetical collapse of wild pollinators on the European continent would worsen micro-nutrient deficiencies, particularly in Europe and to a lesser extent in the rest of the world, with

potential adverse health implications[22,35], which are not addressed within the scope of this study.

## Regional impacts on the European agri-food system

The CAPRI model allows the analysis of regional impacts of a wild pollinator collapse in Europe for 273 regions, most of them being regions at the second level of the Nomenclature of Territorial Units for Statistics (NUTS), henceforth NUTS2 regions. In terms of changes in total crop output, 262 of 273 (or 96%) European regions report a decline, with the highest declines occurring in regions in Austria (Fig. 5A). Total regional crop production in Europe experiences a median decline of 3.9% with the bottom quartile of European regions reporting a decline of 7.2% or greater in magnitude. Output of pollination-dependent crops declines in almost all regions (Fig. 5B). Southern and Eastern European regions, whose agricultural sectors have high dependence on pollination-dependent crops, are affected most severely, particularly Spain, where almost all regions experience declines higher than 20%. Within Northern Europe, the output of

A) Global change in crop output

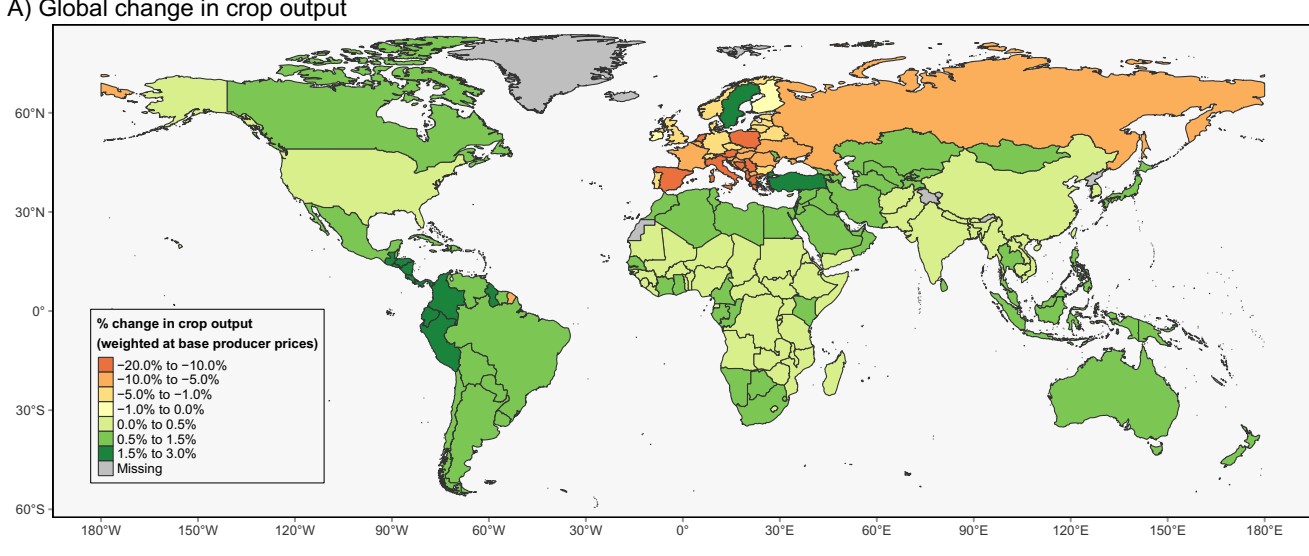

B) Global change in welfare

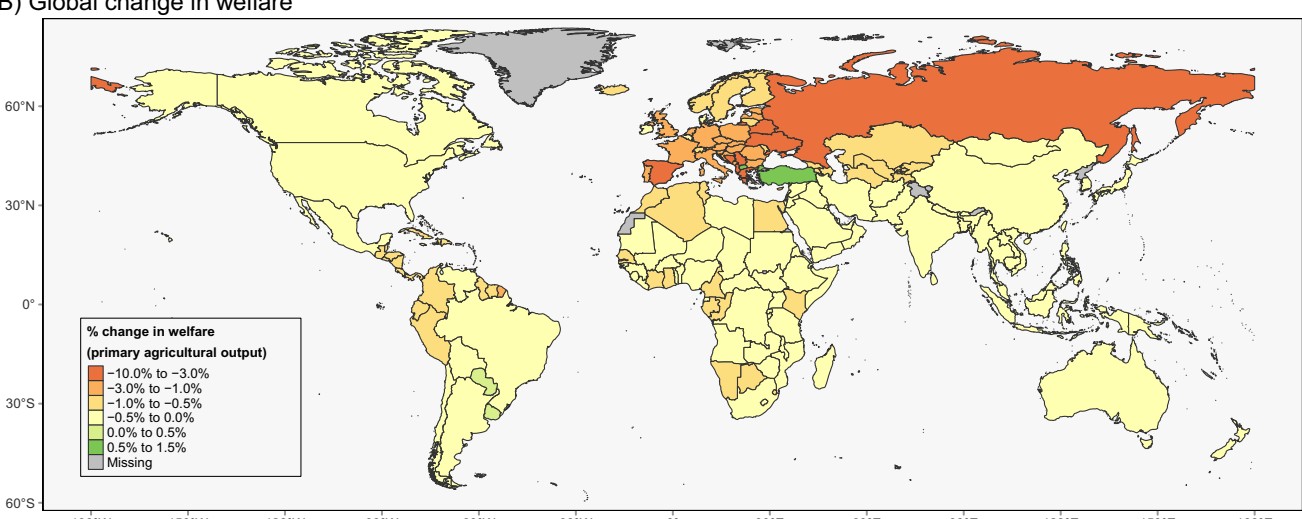

**Fig. 3 | Global propagation of productivity shocks from a European wild pollinator collapse.** Global relative changes in (**A**) crop output and (**B**) welfare resulting from a simulated collapse of wild pollinators in Europe. Results are shown for the scenario using the mean productivity shocks to pollinator-dependent crops in Europe. Changes in crop output are weighted by base producer prices, and welfare changes are expressed as a share of total agricultural output. All values are relative to a baseline scenario without pollinator decline. Results are shown at the country level for all model regions. Source: See Methods for scenario design and Supplementary Materials for source data. © EuroGeographics 2025 for the administrative boundaries[88,89].

pollination-dependent crops experiences a disproportionately high decline in the Netherlands. Six regions in the United Kingdom and Sweden report slight increases, albeit negligible in quantity terms. In parts of Sweden, for example, the reallocation of land to oilseed production results in an overall increase in pollination-dependent crop output. On the median, output of pollination-dependent crops declines by 13.0%, with the bottom quartile of regions showing declines exceeding 17.5%.

The supply of crops that do not require wild pollinators are still indirectly affected because of their cross-price relationships with pollination-dependent crops and due to the re-allocation of production factors such as land. In most regions, pollination-independent crops become more attractive to farmers, since their yield remains stable. This particularly holds for 38 of the 273 regions with marked increases exceeding 1%, particularly in Great Britian, Scandinavia and parts of Eastern Europe (Fig. 5C). Vegetable and permanent crops (fruits and nuts) are the most affected commodity group given their strong dependence on pollination services with a median decline in

production of 10.2%, that is, in roughly half of all NUTS2 regions production falls by more than 10% (Fig. 5D). In the bottom quartile, declines exceed 16.9%.

### Consumer Welfare and Political Economy Implications in Europe

The collapse of wild pollinators would particularly hurt consumers, as food prices would substantially increase while availability decreases. Figure 6A presents the relative consumer welfare declines by each country's gross domestic product (GDP). Within the EU, consumer surplus declines are highest for member states in Eastern Europe, with Romania, Poland, Croatia, and Greece showing the highest relative declines. Interestingly, there is a noticeable overlap between the average voting behavior of parliament members of EU member states on two major biodiversity-related proposals associated with the EU Pollinators initiative[36]: the Nature Restoration Law (Fig. 6B) and the pesticide reduction targets of the Sustainable Use Regulation (Fig. 6C). This voting behavior has a statistically significant negative correlation

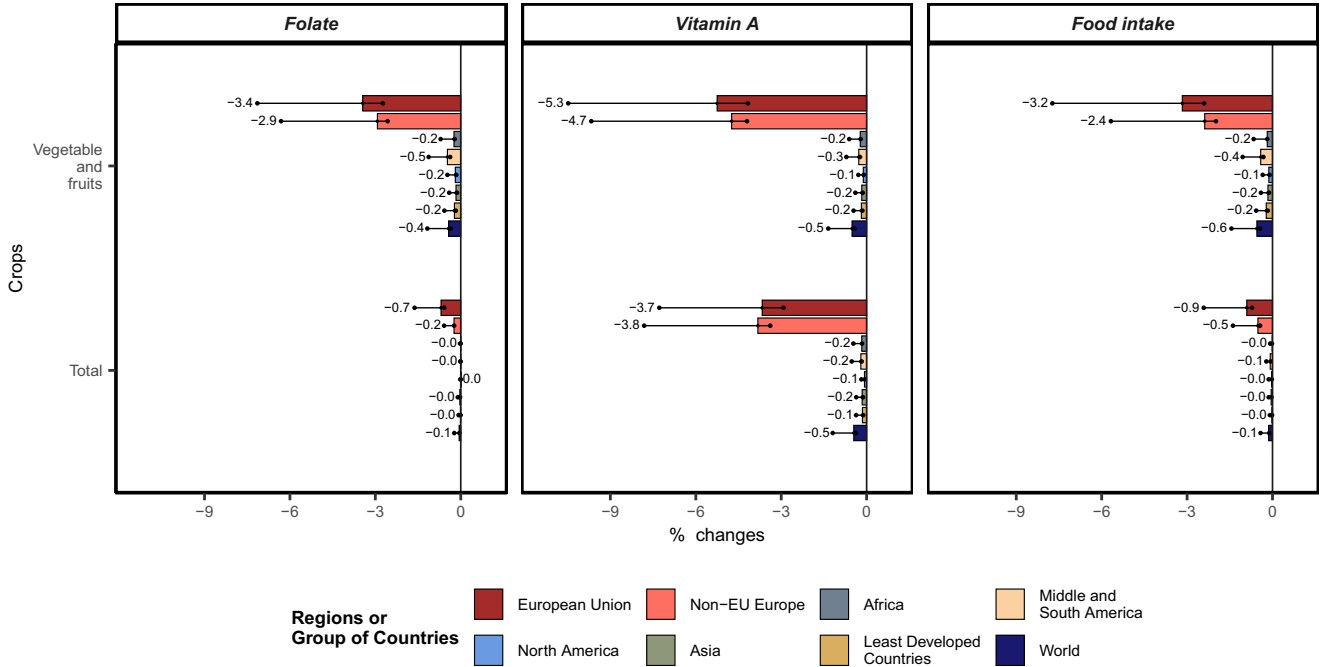

**Fig. 4 | Percentage changes in selected food security indicators across world regions following a simulated collapse of wild pollinators in Europe.** Indicators shown are folate availability, vitamin A availability, and total food intake (in kilocalories per capita and day), aggregated by crop groups (Vegetables and fruits; Total all food). In total, three independent scenario simulations were conducted (*n* = 3). Bars represent the mean outcomes of model simulations using the mean productivity shock scenario. Dots indicate the range of results from two additional model simulations corresponding to the upper and lower bounds of the 95% confidence interval for productivity losses in pollinator-dependent crops. Text labels show the mean result. Values are expressed as percentage changes relative to a baseline scenario without pollinator collapse. See Methods for scenario design and Supplementary Materials for data sources.

with the projected impacts of a wild pollinator collapse on consumer welfare (Fig. 6D), despite the small number of observations. The statistical association is much more pronounced for the Sustainable Use Regulation than for the Nature Restoration Law, where statistical significance is just above the 5% threshold. While the negative correlations do not imply causality, they point to a potential misalignment between ecological vulnerability and political support for biodiversity protection, possibly driven by underlying structural or economic characteristics. Further analysis would be required to disentangle these factors.

Conversely, the positive and statistically significant relationship between changes in producer surplus (as a share of GDP) and voting behavior (Supplementary Fig. 4) suggests that potential economic benefits, like price increases due to scarcity, might indirectly shape policy preferences. However, it is unlikely that producers are explicitly factoring in the possible impacts of a wild pollinator collapse in their current decision-making or exertion of political influence. We also need to consider that producer surplus changes relative to GDP are related to overall agricultural reliance, reflecting both stronger economic exposure to pollinator decline and possibly a higher concentration of producer interests. This concentration may, in turn, translate into political resistance toward biodiversity conservation initiatives perceived as economically restrictive.

### Sensitivity analysis
We test the robustness of the model results through a systematic sensitivity analysis of key model and scenario parameters (Supplementary Methods). Generally, results outside of the EU exhibit low sensitivity to parameter variations (Fig. 7), while outcomes within the EU are particularly responsive to changes in parameters related to yield dependence on pollination services and the contribution of wild pollinators. This is expected, because these biophysical parameters

directly determine the magnitude of the productivity shocks of a collapse in wild pollinators. Conversely, the results for the EU show minimal sensitivity to adjustments in supply elasticity parameter values. The combination of parameters that yields the largest welfare decline—namely, high wild pollinator contribution, high yield dependence on pollination services, low supply elasticity, and high trade elasticities—leads to a welfare reduction of 3.5% within the EU's agri-food system, equivalent to 0.2% of GDP. This is approximately twice the magnitude of the welfare decline observed under the core scenario (1.7% welfare decline, equivalent to 0.1% of GDP). Domestic production experiences less decline when assuming a lower degree of substitutability between foreign and domestic goods by reducing the Armington trade elasticity values. However, this adjustment results in higher prices, benefiting local producers but negatively impacting consumer surplus. Consequently, while the decline in consumer surplus becomes more pronounced, the overall welfare decline is moderated due to a compensating increase in producer surplus.

### Discussion
This study examines the large-scale consequences of a wild pollinator collapse across Europe, utilizing the latest data on crop yield dependence and the contributions of wild versus managed pollinators. By applying the CAPRI model—a detailed global agriculture partial equilibrium modeling system—we capture the potential impacts on commodities and regions, highlighting CAPRI's strengths in commodity and regional resolution as well as its capacity to depict long-term market adjustments through shifts in land use, production, trade, and consumption. Our findings indicate that a wild pollinator collapse in Europe would seriously disrupt crop production, trade flows, and food security. While European producers and consumers would bear the brunt of the impact, trade-induced spillover effects would reach the global agri-food system, with a projected global economic welfare loss

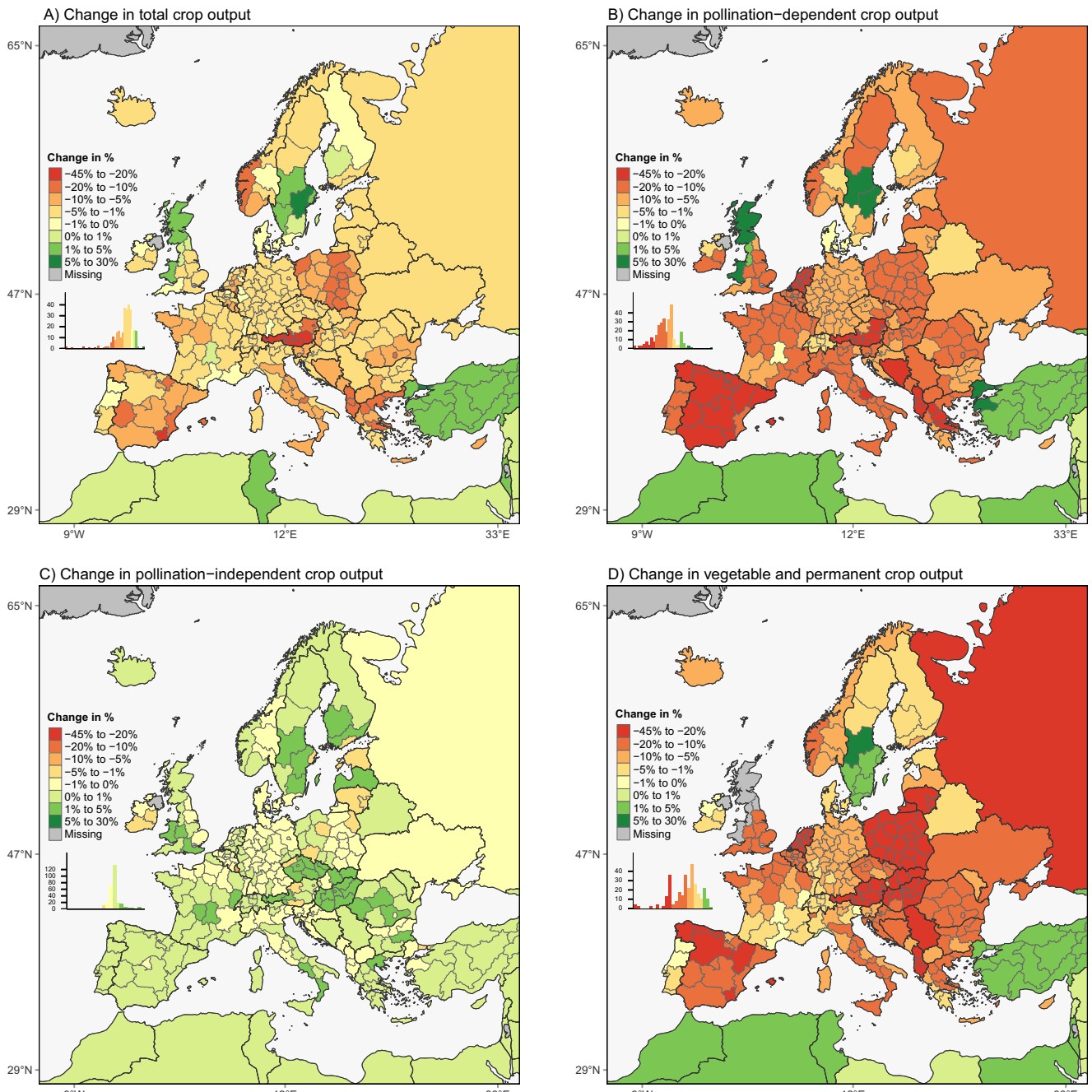

**Fig. 5 | Relative changes in agricultural output in Europe resulting from the mean productivity shocks caused by a wild pollinator collapse in Europe.** Panel (**A**) shows the changes in total crops; (**B**) pollination-dependent crops; (**C**) pollination independent crops; and (**D**) vegetable and permanent crop production. All changes are expressed relative to a baseline scenario without pollinator loss.

Results are based on CAPRI model simulations and presented at the national NUTS1 or subnational NUTS2 level where available. Source: See Methods for scenario design and Supplementary Materials for source data. © EuroGeographics 2025 for the administrative boundaries[88,89].

of €34.4 billion annually. Welfare losses within Europe and the European Union would amount to €23.8 billion and €12.4 billion, respectively, translating to €557 and €724 per hectare of cropland dependent on wild pollinators. Simulating the same scenario with the alternative, but older data on dependence ratios from Klein et al.[17] would result in 20.6% lower global economic welfare losses of €27.3 billion (Supplementary Table 2), reflecting the mostly lower dependence ratios of this data compared to the most recent data provided by Siopa et al.[20] (see difference in productivity shocks in Supplementary Table 3).

This study sheds light on the effects of an asymmetric biodiversity crisis, like the collapse of wild pollinators in one region, on

international trade and adaptation responses through cross-price relationships and land-use adjustments. Globally, and particularly within Europe, the model projects cropland expansion that increases pressures on biodiversity[6,13] and potentially triggers negative feedback loops that affect other ecosystem services, such as natural pest control[37]. Although European countries could offset production losses through imports, this approach could limit nutrient-dense food availability in other regions, notably affecting global fruit and vegetable supplies (Fig. 4) and hindering transitions toward sustainable, healthier diets[38]. This study does not directly address the health impacts of these dietary effects, although other research[39] has highlighted

A) Change in consumer welfare as a share of national GDP in 2017
following a collapse of wild pollinators in Europe

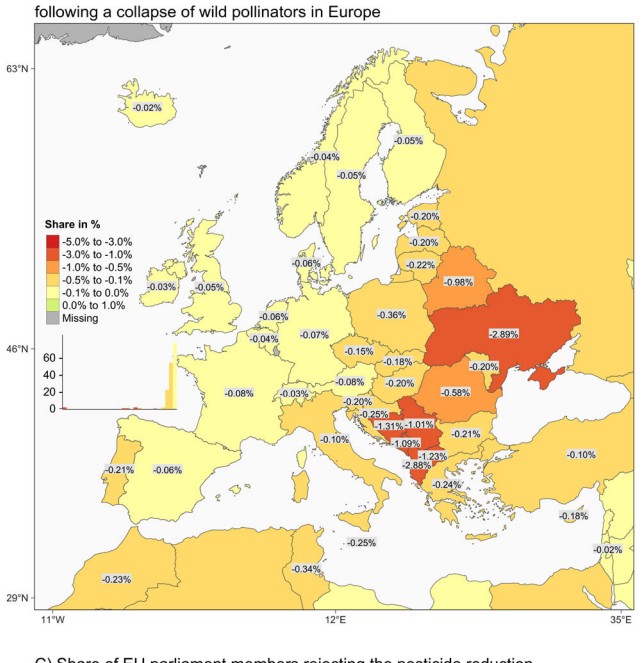

B) Share of EU parliament members rejecting the Nature Restoration Law
on 12.07.2023

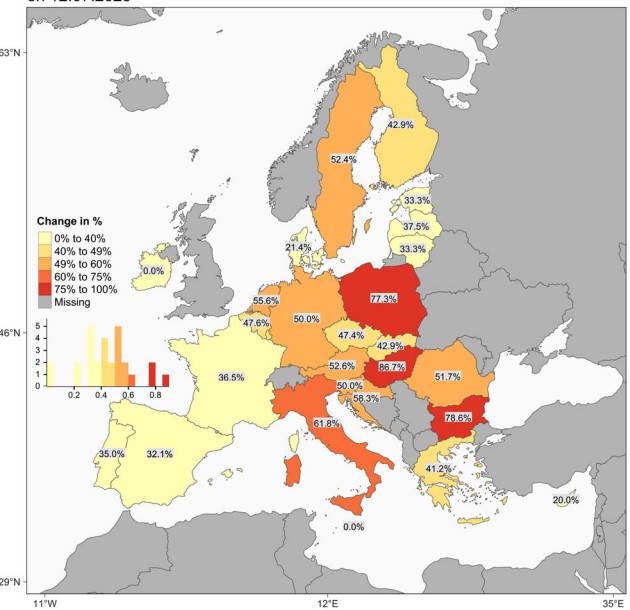

C) Share of EU parliament members rejecting the pesticide reduction
targets of the Sustainable Use Regulation on 22.11.2023

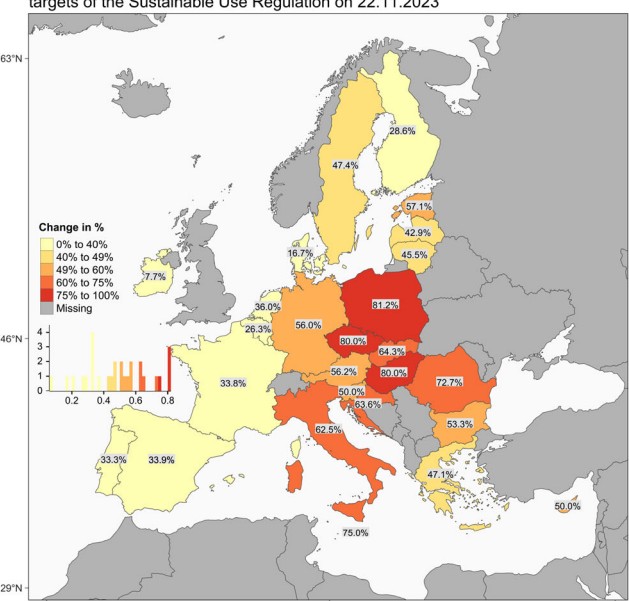

D) Consumer welfare changes as a share of national GDP vs. share of EU
parliament members rejecting 'pollinator-friendly' policy proposals in 2023

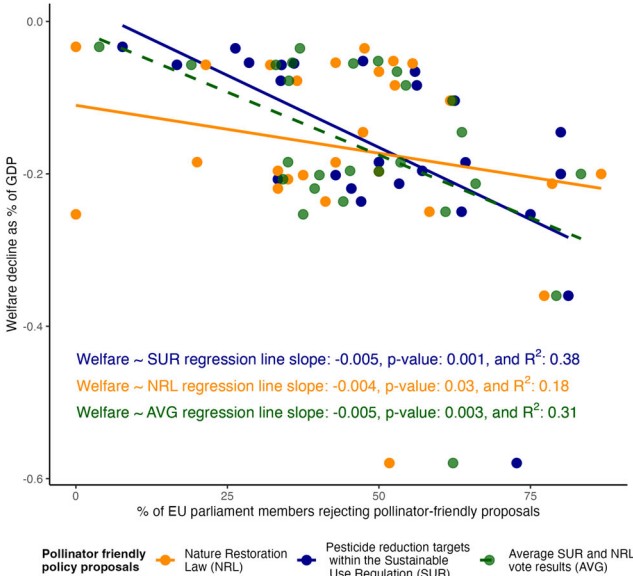

**Fig. 6 | Association between consumer welfare changes and voting behavior on biodiversity-friendly policy proposals in the European Union.** Panel (**A**) shows changes in consumer welfare as a share of national gross domestic product (GDP) resulting from the mean productivity shocks caused by a wild pollinator collapse in Europe. Panel (**B**) shows the share of EU parliament members rejecting the Nature Restoration Law (NRL) during the vote on 12.07.2023. Panel (**C**) shows the share of EU parliament members rejecting pesticide reduction target proposal (A9-0339/2023) under the Sustainable Use Regulation (SUR) on 22.11.2023. Panel (**D**) presents

a scatter plot and population-weighted regression analysis of the relationship between consumer welfare changes and shares of EU parliament members rejecting pollinator-friendly policy proposals. Regression lines are shown for NRL, SUR, and their average (AVG). Voting data are based on 26 EU member states ($n = 26$); Luxembourg is included within Belgium in the CAPRI model. Source: See Methods for scenario design and Supplementary Materials for source data. Voting data from the European Parliament[90,91]. © EuroGeographics 2025 for the administrative boundaries[88,89].

potential health implications (although without considering land-use, production, trade, or consumption adjustments).

Our welfare and food intake outcomes represent average national households, limiting the ability to analyze impacts on particularly vulnerable groups. This is a known constraint in multi-regional or global economic simulations. Nonetheless, our findings reveal notable disparities in welfare decline across Europe (Fig. 3B), with low-income countries, where food expenses consume a larger share of household

budgets, facing disproportionately high impacts. This likely extends to lower-income households within wealthier countries. The production of pollination-dependent crops, such as oilseeds, vegetables, and fruits, is vital for farmers and rural communities[27], though the economic effects of a pollinator loss on these communities are complex. Our model suggests that some producers of pollination-dependent crops may benefit from the "King-Davenant law"[24], as higher prices can offset yield reductions for certain crops like sunflower, tomatoes, and

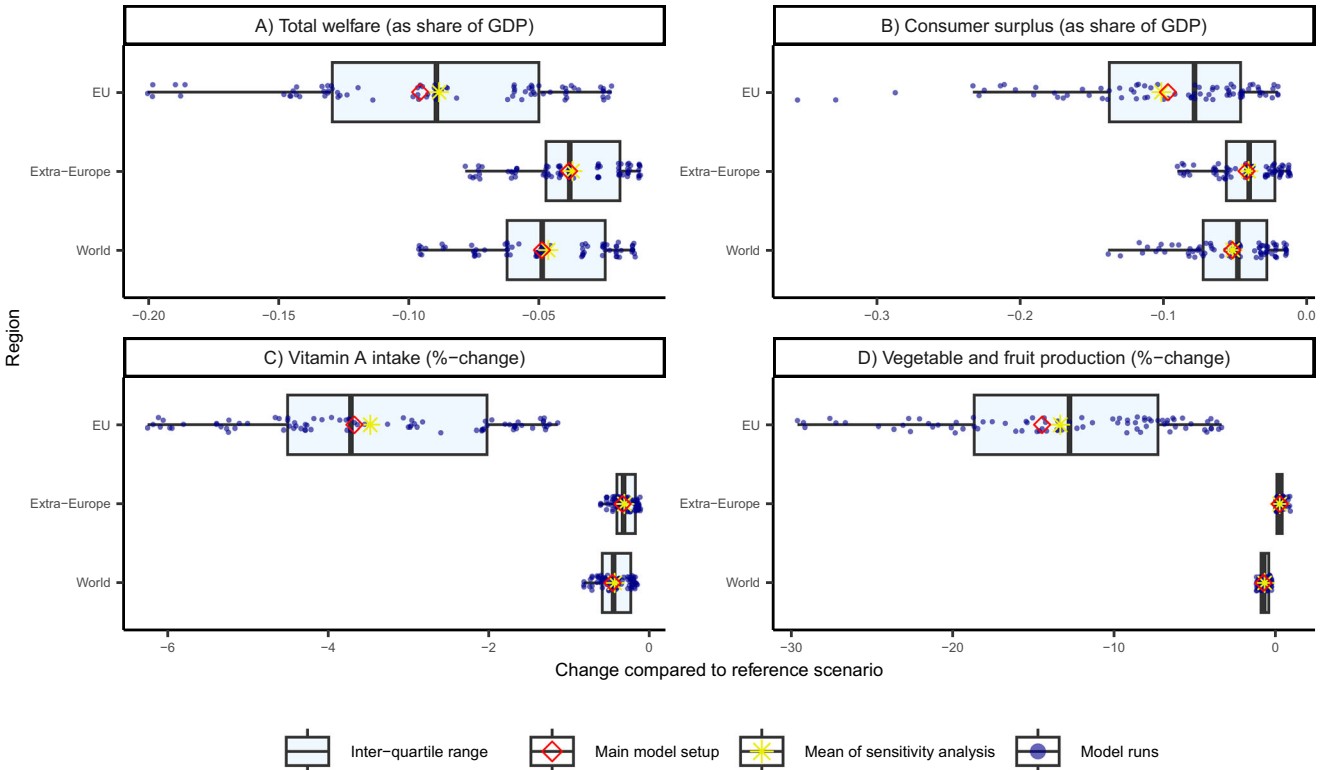

**Fig. 7 | Sensitivity analysis of main model results under variation of key model parameters.** Each dot represents the outcome of one of 81 model runs ($n = 81$), generated by systematically varying selected model assumptions. The red diamond marks the result of the main model setup, while the yellow star denotes the mean result across all sensitivity runs. The boxplot summarizes the distribution of model results: the box spans the interquartile range (IQR), the central vertical line indicates the median, and the whiskers extend to the smallest and largest values within 1.5 times the IQR. Panels show changes in (**A**) total welfare, (**B**) consumer surplus, (**C**) vitamin A intake and (**D**) vegetable and fruit production across the EU, Extra-Europe, and the world. Extra-Europe refers to all regions outside of Europe. Source: See Supplementary Materials for source data.

apples. Lower yields per area could increase harvest costs[23,24], which are not captured in detail in our model. Therefore, the changes in producer surplus need to be interpreted with caution.

The significant correlation between consumer surplus losses and EU member states' resistance to pollinator-friendly policies, measured by the average voting behavior of EU parliament members, is an incidental finding that warrants further investigation. This relationship may suggest that regions facing the greatest economic impacts from a pollinator collapse may be less supportive of biodiversity policies, potentially due to concerns about immediate economic or other reasons, such as doubts about the policies' effectiveness[40,41]. A similar pattern was observed in a study that showed that countries in the Americas and Asia, despite their increasing reliance on pollination-dependent crops, often expand agricultural practices that undermine pollination services[42]. Our study did not originally set out to examine any political-economic aspects, and it remains limited in scope without investigating causal relationships. Future research could investigate potential confounding factors, such as countries' agricultural dependency as a share of GDP, to better understand why some regions may exhibit higher resistance despite clear economic risks. Additionally, understanding how changes in both consumer and producer surplus interact with policy preferences could offer valuable insights into the socio-economic dynamics of environmental policy support. For instance, while consumer losses correlate negatively with policy support, the positive correlation between producer surplus gains and voting behavior may reflect different economic incentives or concerns about costs (Supplementary Fig. 4).

Greater awareness of the value of ecosystem services from wild pollinators could strengthen conservation efforts, particularly in those regions that are potentially most affected by adverse changes in pollinator populations. Such measures might include subsidies for low-input or organic farming, crop diversity, and the establishment of semi-natural habitats, such as hedgerows or wildflower strips[42–44]. Current research estimates that 20–25% of agricultural and human-modified landscapes need to be dedicated to natural or semi-natural habitats to maintain biodiversity's capacity to provide ecosystem services[45,46]. The estimated welfare loss of €12.4 billion in the EU could justify annual spending of €633 per hectare on biodiversity-friendly measures across 20% of Europe's arable land, aligning with Germany's average subsidy of €650 per hectare for wildflower strips[47]. Most other agro-environmental programs promoting biodiversity require less funding[24], though farmers' reluctance to adopt these measures remains a challenge[48]. Notably, many conservation measures benefit only the subset of wild pollinators that frequent agricultural fields, while the costs of protecting endangered species may not align with their contributions to ecosystem services[49].

The welfare losses reported in this study serve as an indicator of the value of wild pollinators. However, these estimates rely on specific methods and assumptions[50]. For example, our model assumes that managed pollinator populations remain stable despite the wild pollinator collapse. While the relative contribution of wild and managed pollinators to crop pollination clearly varies across crops (see Table 1) and regions, global evidence suggests that, on average, both groups contribute approximately equally to overall pollination services[19].

Under the strong and simplifying assumption of perfect substitutability between managed and wild pollinators, the loss could theoretically be mitigated by nearly doubling Europe's 25 million honeybee colonies from 2017, which would cost between €2.1 billion and €3.6 billion annually, or approximately €49.7 to €83.4 per hectare for pollination-dependent crops (see Supplementary Methods). While

such an approach is consistent with valuation techniques such as beehive rental or replacement cost models[50–53], it excludes the costs of scaling up other managed pollinators like bumblebees and neglects the likely rise in marginal costs of delivering pollination services. Moreover, the assumption of equal contribution between wild and managed pollinators, although supported by global averages[19], must be treated with caution due to substantial spatial and crop-specific variability. In many cases, particularly for crops highly dependent on specific wild pollinator species, substitution is either limited or infeasible[18,19,54,55]. Therefore, this scenario cannot serve as a realistic estimate of welfare costs and is presented here for illustrative purposes only.

Beyond market impacts, wild pollinators provide important non-market values, including cultural, aesthetic, and intrinsic values[44,54]. These values are typically assessed through willingness-to-pay estimates[55–57]. Recent data suggest that German households are willing to pay €227 to €447 per year for wild bee conservation[56]. To be conservative and to account for variations in income, cultural values, and ecological awareness across the EU, we assume that half of this lower bound, €114 per household annually, applies to the EU's 193 million households as of 2017. This results in an estimated annual value of €21.9 billion for wild pollinators – nearly double the EU's estimated welfare losses of €12.4 billion from our study. While this extrapolation already discounts the lower bound willingness-to-pay, it may still be an overestimation due to hypothetical bias or embedding effects, where respondents consider wild bee conservation as part of broader nature conservation goals[56].

Several limitations affect our model and simulations. First, we simulate a complete collapse of wild pollinators, representing a hypothetical and extreme scenario, despite mounting evidence of drastic insect declines in Europe and globally[1,15,29].

We define a wild pollinator collapse as a 90% decline in population over the model's time horizon of 13 years (2017 to 2030), corresponding to an annual decline rate of 15.2%. There is a notable lack of comprehensive data on pollinator population trends in Europe. The available evidence suggests annual rates of decline ranging from 0.8% to 14.1%, with an average of 5.8% (Supplementary Methods). These estimates vary widely in terms of taxa, geographic coverage, and methodology. They include studies on hoverfly (Syrphidae) abundance[1,58] and bee and hoverfly occupancy[59,60] - both key pollinator taxa[18] – as well as studies on trends in the abundance or biomass of broader groups such as flying insects[2,61], arthropods[29], and terrestrial insects[30]. Extrapolating these trends over a period of 13 years from 2017 to 2030 in the absence of mitigation efforts indicates that a wild pollinator collapse in Europe, as defined in our study, is unlikely but cannot be entirely ruled out (Supplementary Methods). Although extreme, investigating this "what if" scenario provides valuable insights into how severe pollination service declines might impact regions, markets, and stakeholders, informing biodiversity conservation strategies, land-use planning, and agricultural policy. Additionally, it highlights the urgent need for spatially comprehensive, long-term monitoring of wild pollinators in Europe to better assess the plausibility of such extreme scenarios.

While our model considers adaptation through land-use and trade adjustments, a further limitation is that it does not include technical substitutions, such as hand pollination[62] or robotic pollinators[63]. However, there is still insufficient data about the cost of these mitigation strategies and particularly robotic pollinators, if ever technically and economic feasible, could cause further ecological and moral risks[64]. Our estimates for crop dependence on pollination vary based on conservative assumptions, modeling the lower and upper bounds for productivity shocks, and conducting sensitivity analyses. Dependence ratios come from studies with differing methodologies (e.g., variations in the quality of "open pollination" references) and often overlook variations in crop quality and variety[20,65,66]. This study

conservatively excludes potential crop quality deterioration and wild pollinator contributions to seed production[67], a factor crucial for forage crops and vegetables that accounts for up to 25% of total pollination service value[68].

Finally, the ecological and social costs of a wild pollinator collapse extend beyond agriculture, potentially disrupting ecological processes, food webs, and overall biodiversity resilience[9,44]. Wild pollinators are essential for the reproduction of most wild plant species and their diversity[69–71]. These wild plants provide food for granivorous species, while wild pollinators themselves serve as a food source for higher trophic levels[3,12]. Hence, although the agricultural impacts can be monetized, the broader, interconnected ecological losses remain invaluable for long-term stability.

This study highlights the far-reaching consequences of a wild pollinator collapse on Europe's agri-food system while accounting for adjustments via land-use, production, trade, and consumption changes. The loss of wild pollinators would drastically reduce crop yields, disrupt trade, and diminish food security, particularly impacting nutrient-rich foods like fruits and vegetables. While European countries may offset some losses through increased imports, this adaptation would have repercussions for food availability and security globally, potentially exacerbating global inequalities and undermining the efforts to make diets healthier and less reliant on animal-based products[38,72,73].

Protecting pollinators remains high on the agenda of many governments, particularly within the European Union. At the same time, resistance to biodiversity conservation measures persists. Our results reveal a pronounced regional disparity in the projected economic impacts of a wild pollinator collapse, with several EU member states facing disproportionately high consumer welfare losses. Some of these same countries have shown lower levels of parliamentary support for recent EU biodiversity-related legislation. While this correlation is not causal, it suggests a potential misalignment between ecological vulnerability and political positioning. A key implication for EU institutions may be to design biodiversity and pollinator-friendly policies in a way that better aligns ecological risk exposure with the perceived benefits of such policies, thereby helping to build broader political coalitions for biodiversity protection.

## Methods
We employ the CAPRI model to analyze the counterfactual effects of a collapse in all wild pollinators on the European continent. CAPRI is a partial equilibrium model of the agri-food system[74] and is built on two model components: a global market module and a regionally disaggregated supply module for Europe mostly at NUTS2 level.

The global market module captures supply, demand (human consumption, feed, and processing), and trade for a total of about 56 agricultural commodities[75,76]. Markets clear in physical terms according to behavioral demand and supply functions[74]. The global market covers about 40 countries, trade blocs or country aggregates, and trade flows are modelled according to the Armington assumption[77]. Hereafter, we refer to them as CAPRI regions. Commodity markets and prices are connected via cross-price linkages. A price transmission function reflects price wedges arising from factors such as import tariffs or export subsidies, and thereby links prices across regions[78].

The supply module captures a high level of detail of the European agricultural sector by disaggregating the supply into about 273 NUTS2 regions. Farming decisions in each region are modelled using a non-linear positive mathematical programming approach. In each region production activities for the main crop and livestock products are further disaggregated into a low- and high-intensity variant[75]. Each NUTS2 region can therefore be thought of as a "representative regional farm supply model" where supply of crops and animal outputs is modelled according to the maximization of an aggregated profit function subject to various constraints[78]. These constraints include the

availability of production resources (e.g., labor and land), policy restrictions (production quota, set aside obligations), and restrictions due to feeding requirements.

The CAPRI model reaches an equilibrium by running both modules sequentially until convergence between both modules is reached. Within each iteration, the market module feeds in commodity prices into the supply module, while the supply module updates the supply and feed demand within the market module[78]. Thereby, agricultural demand, production, trade, and market-clearing prices are determined consistently both at the global and detailed, regional level. This is one of CAPRI's strengths, particularly regarding the implications of policy or other exogenous changes on the EU-level on global agricultural markets and vice versa. In addition to the endogenous model mechanisms, CAPRI also performs various ex-post calculations to determine associated environmental emissions and other biophysical results. To date, CAPRI has been widely applied for ex-ante assessments of economic and environmental impacts of exogenous shocks such as climate change, diet shifts, and policy changes[37,38,75,79–83].

## Scenario formulation

The model scenario assumes that the total wild pollinator population on the European continent will have collapsed over the model's 2017-2030 time horizon (14 years). We define a collapse as a decline of at least 90% in the wild pollinator population by 2030. A similar threshold is used in other studies to define collapses in the biomass of fish stocks[84]. This deliberative "what-if" scenario allows us to assess the region-specific agricultural sector's dependency on wild pollinators, including the role of market adjustments in the global agri-food system. This hypothetical scenario is built on a recent dataset of dependence ratios, i.e., the degree to which crop yields are determined by animal-mediated pollination services[20] and on information on the extent to which pollination services are provided by wild or managed pollinators[19]. Modelling the collapse of wild pollinators on the European continent goes beyond the existing approaches that have previously modelled total (and global) collapse of both managed and wild pollinators. As Bauer and Wing[26] point out, the literature provides little or no information on the substitutability of animal-mediated pollination services with other market inputs (e.g., manual pollination). Consequently, and analogous to Bauer and Wing[26], we implement the collapse in pollinators as a productivity shock using dependence ratios. However, our approach explicitly differentiates between the contributions of wild and managed pollinators. In the model itself, we also explicitly assume that there is no change in the population of managed pollinators, which would reflect a zero-substitution elasticity. Since, depending on the crop and location, managed pollinators can partially substitute wild pollinators, we explore an alternative scenario in which all wild pollinators could be substituted (see Discussion).

We utilize data from the latest comprehensive review on the dependence of crop production for human consumption on animal pollination[20]. This review covers all relevant FAO crops and finds that at least 35% of global crop production depends on animal-mediated pollination at varying degrees, expressed by dependence ratios. In comparison with the Klein et al.[17] data, the dependence ratios reported by Siopa et al.[20] are substantially more differentiated. For example, Klein et al. report for apples as well as peaches and nectarines a "great dependence ratio" equal to 65%, while the mean dependence ratio for apples reported by Siopa et al. is 73%, but only 37% for peaches and nectarines. For a few crops, Siopa et al. report a dependence ratio of 100% (e.g., for quinces or pumpkins), which for technical reasons, are implemented as a 99.9% productivity shock. Each FAO crop and its dependence ratio are mapped to the CAPRI model commodity structure. Reilly et al.[19] measured the relative contribution of wild pollinators towards four different yield indicators (fruit weight ($n = 17$), yield ($n = 35$), fruit count ($n = 6$), and fruit set percentage ($n = 35$)) using data

on 32 pollination-dependent crops from 93 studies and 31 countries. Whether considering the global data or just the data from European sites, wild and managed pollinators are found to contribute equally to pollination-dependent crop yields. Yet, wild pollinators' relative contribution varies strongly across crops (see also Table 1). The data from European studies on the relative contribution of wild pollinator per crop is mapped to the CAPRI model commodity structure. Given the lack of data, the average relative contribution was assumed for the crops "Tomatoes" and "Soybeans", which are also separate commodities in the CAPRI model.

The productivity shocks are derived for each of the ten pollination-dependent commodities that are included in CAPRI and produced in Europe. For those commodities that do not refer to a single but multiple crops (e.g., commodities like "Other Vegetables" or "Other Fruits"), the productivity shock is determined by the region-specific production value shares of the relevant pollination-dependent crops in the respective commodity.

The productivity shock, $\delta_{c,n}$, for a commodity $c$ and country (or region) $n$, is calculated as:

$$\delta_{c,n} = 1 - \sum_{i=1}^{I} \text{shrcom}_{c,i,n}\, \varphi_c\, D_i$$

where $\varphi_c$ is the relative contribution of wild pollinators, $D_i$ denotes the overall dependence ratio of crop $i$ on pollination services provided by both managed and wild pollinators, and $\text{shrcom}_{c,i,n}$ is the share of crop $i$ in total production value of a CAPRI commodity $c$ in country $n$. The shocks are identical across countries if the CAPRI agricultural commodity is directly mapped to a single crop included in the Siopa et al. data. This applies for four commodities: Rapeseed, Sunflower, Soya, and Tomatoes. For the remaining six pollination-dependent commodities, the shocks are country-specific because multiple crops are aggregated in one commodity. The aggregation shares rely on information on production value taken from the FAOSTAT database[85] for the model's base year 2017. The calculation of the productivity shocks was performed using the GAMS software, for which the code is included in the electronic supplementary materials.

The computed productivity shocks, $\delta_{c,n}$, aggregated for Europe are reported in Table 1. The table also reports the dependence ratio on managed and wild pollination services as well as the relative contribution of wild pollinators. A 95% confidence interval of the productivity shocks is estimated assuming a triangular distribution and using the minimum and maximum values reported for $D_{n,i}$. The shocks specific to each CAPRI region are reported in the Supplementary Table 3 for the main scenario based on the dependence ratios from Siopa et al.[20], but also for an alternatively conducted scenario based on the older data from Klein et al.[17].

## Scenario implementation and analysis

The estimated mean productivity shocks and their 95% confidence interval are implemented for all European CAPRI regions (see Supplementary Code 1). Turkey is considered as part of the Asian continent, and its NUTS2 regions are therefore not affected by the wild pollinator collapse. The simulation results from the wild pollinator collapse scenario are compared against a reference scenario, which simulates the "business-as-usual" developments within the agricultural sector based on projections on demographic and economic growth, as well as projected changes in agricultural policy and markets. It is largely derived from the OECD-FAO Agricultural Outlook[86], but also considers historical trends, expert information, and other sources[74]. The scenario is implemented by activating both the supply and market module in CAPRI. The year 2017 is calibrated as the model's base year. The EU composition excludes Great Britain, and the final time horizon of the simulation is the year 2030, by which wild pollinator populations are assumed to have collapsed. The model's agents (farmers,

processors, traders, consumers, etc.) are assumed to have full knowledge about the impending collapse and thus adjust their decisions accordingly.

A key indicator for the scenario analysis is changes in societal welfare. Total welfare change is measured as the sum of changes in consumer surplus, producer surplus of farms, rents for feed producers and dairy and agro-processors, and finally, government revenues from tariffs and tariff rate quota rents and expenditures from subsidies. Consumer surplus changes are measured as the income a consumer would need to arrive at the same utility level as in the reference scenario. Producer surplus changes are changes in the scenario's farm profits compared to farm profits in the reference scenario. More details on the welfare calculation are provided in the model documentation[76].

## Reporting summary
Further information on research design is available in the Nature Portfolio Reporting Summary linked to this article.

## Data availability
Country-specific data on crop production, cultivated area, and producer prices are publicly available from the FAOSTAT database (https://www.fao.org/faostat/en/#data). The most recent crop-specific dependence ratio data are available from the supplementary materials of Siopa et al. (2024) (https://besjournals.onlinelibrary.wiley.com/doi/full/10.1111/1365-2664.14634). Alternative, but older, crop-specific dependence ratio data from Klein et al. (2007) are available at https://royalsocietypublishing.org/doi/10.1098/rspb.2006.3721. Data on wild pollinator contributions are available from the supplementary materials of Reilly et al. (2024) (https://onlinelibrary.wiley.com/doi/10.1111/geb.13843). Source data to replicate the figures in the main article and Supplementary Information are provided as separate Supplementary Data files. The data and code to derive the three productivity shock scenarios (mean and 95% confidence interval) are available on the GitHub depository (https://github.com/ArndtFeuerbacher/WPC_Shocks) and are archived on Zenodo (https://doi.org/10.5281/zenodo.16847817). The model results generated and visualized in this study are provided in the Supplementary Information/Source Data file. The data used to derive the productivity shocks are available at https://github.com/ArndtFeuerbacher/WPC_Shocks and are archived on Zenodo (https://doi.org/10.5281/zenodo.16847817). The CAPRI model database can be downloaded from www.capri-model.com and is also available from the authors upon request. Source data are provided with this paper.

## Code availability
The code to derive the three productivity shock scenarios (mean and 95% confidence interval) is available on GitHub (https://github.com/ArndtFeuerbacher/WPC_Shocks) and archived on Zenodo (https://doi.org/10.5281/zenodo.16847817). The CAPRI model code used for this analysis (Trunk version−September 2022) is open source and freely available for download at www.capri-model.com. It is also available upon request from the authors. The Supplementary Code 1 contain the batch file, GAMS code, and productivity shocks needed to implement the scenarios within the CAPRI model. Data and result analysis were conducted using R (v.4.4.1).

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

## Acknowledgements

The authors thank three anonymous reviewers for their valuable comments on earlier versions of this study, as well as all of the CAPRI model developers and contributors. Arndt Feuerbacher is grateful to Christian Lippert, Anne-Christine Mupepele, Ingo Grass, Thomas Fellmann, Tim Williams, Franziska Steinhübel, and participants at the 16th congress of the European Association of Agricultural Economists (EAAE) for stimulating discussions, insightful feedback, and their encouragement. Any remaining errors and omissions are the sole responsibility of the author team. A.F. was financed through the "Add-on Fellowship for Interdisciplinary Economics and Interdisciplinary Business Administration" by the Joachim Herz Foundation and the junior research group "BEATLE" funded by the German Federal Ministry of Research, Technology and Space (BMFTR) under its Social-Ecological Research funding priority (funding no. 01UU2201A).

## Author contributions

A.F.: Conceptualization, Methodology, Software, Formal Analysis, Investigation, Writing–original draft, Writing– review & editing, Visualization; M.K.: Methodology, Software; J.L.M.S.: Conceptualization, Writing– review & editing; C. W.: Conceptualization, Writing– review & editing.

## Funding

## Competing interests

The authors declare no competing interests.
