## [Transparent Peer Review file · Nature Communications]

The Economic, Agricultural, and Food Security Repercussions of a Wild Pollinator Collapse in Europe

Corresponding Author: Professor Arndt Feuerbacher

Version 0:

Reviewer comments:

Reviewer #1

(Remarks to the Author)

Thank you for the invitation to review this article. In this paper, the authors model a total global collapse of both managed and wild pollinators on food production, land use and nutrient consumption. They assume no possible substitution to other forms of pollination (hand or mechanical) and take the global pollination dependency ratio as fixed, based on Klein et al (2007). The innovation of this paper is that they use a detailed partial equilibrium global agricultural model (CAPRI) to simulate the effect of this shock on food prices and consumption, allowing for changes in intensification, cropped area, international trade and crop substitution to mitigate the production shock. Because their model allows for these mitigating effects, their results show smaller economic costs than earlier work. Even so, their results highlight the wide-ranging effects of such a loss in pollinators, inducing a 6.1% increase in land cultivation, a shift to grains and meats and a 0.4% decrease in global GDP.

The benefit of the paper is that by using a global model that has relatively detailed agricultural product classifications, they are better able to capture the detailed pollination dependence by crop. That said, given that pollination dependent crops often have relatively non-pollination dependent alternatives, their still relatively broad crop classifications will miss out on some likely substitutions (pistachios for almonds for example).

They acknowledge that the assumption that both wild and managed pollinators will be wiped out is quite strong. I appreciate that they do not have data to model the decline of only wild pollinators except for some crops, but this is a large enough assumption that it might warrant some alternatives as robustness tests. For example, instead of productivity loss, they might consider using the additional costs of increasing the number of managed pollinators as a lower bound.

One aspect of their model that was not perfectly clear was that farmers are allowed to mitigate the loss in productivity from pollination by intensifying production. As they note, this mechanism is particularly important for sub-Saharan Africa. What was not clear was the assumptions made in the model about the production function and how other inputs interact with the loss of pollination services. This relationship might be quite different for different crops.

By construction, the paper ignores other ecological costs of pollinator loss such as their effect on non-cultivated plants and as part of a food chain. Further, earlier work shows that natural areas serve as wild pollinator habitat, so that farmland expansion from a reduction in pollinators might trigger a negative feedback loop. Similar threats to pollinators may arise from intensification itself. Assuming a partial pollinator decline, some discussion of these other effects might be useful (e.g. Potts et al 2016).

Last, I struggled a bit in terms of the story the authors are trying to tell. I appreciated the authors' motivation based on specific legislative attempts to preserve biodiversity and wild pollinators in Europe, and the findings that some of the locations most severely hurt by pollinator loss are those areas that do not support these interventions. The simulation they run, however, is a global collapse of pollinators, where the link to EU specific legislation is less clear. They might consider instead a narrower simulation of the effect of a EU pollinator collapse on global agricultural production and consumption. Alternatively, they might consider how a global collapse might be mediated by pollinator preservation in the EU. Or they might cite efforts in other locations to protect pollinator habitat to look at global costs and benefits of such efforts.

In summary, this paper contributes to our understanding of the importance of pollinators to the global agri-food system by modeling the effect of a production shock to pollination-dependent crops, allowing for farmers and consumers to respond by

changing crops, production practices, land area and consumption. They also highlight the regional distribution of the impact of these changes. The drawback of the article is that the authors assume a complete loss of pollination, no change in crop breeding or other measures that could reduce pollination dependence, and by the nature of a large global model of agricultural production, still have relatively broad crop categories that may mask further substitution effects. I am uncertain whether this contribution is sufficient for Nature Communications, but I do believe that the paper makes valuable contributions to the literature on pollinator impact.

Potts, S., Imperatriz-Fonseca, V., Ngo, H. et al. Safeguarding pollinators and their values to human well-being. *Nature* 540, 220–229 (2016). <https://doi-org.proxy.library.ucsb.edu/10.1038/nature20588>

(Remarks on code availability)

Reviewer #2

(Remarks to the Author)

The paper “Impacts of Pollinator Declines on Agri-Food Systems in Europe and Beyond”, assesses the possible effects on agri-food markets of a worldwide collapse of pollinators by 2030. For the assessment, the authors rely on estimates on pollination dependency ratios from Klein et al. (2007), and they apply respective yield shocks in the CAPRI model.

Major Comments:

1. Novelty and Transformation:

In general, the approach of using pollinator-related yield shocks of another study into the partial equilibrium model CAPRI is sound. However, the major concern with the presented work is that, in my opinion, it lacks novelty and a major breakthrough as it uses estimates on pollination dependency from a study in 2007 and inserts them into an existing partial equilibrium model without any model improvements. The authors fail to highlight in the text why this work provides a major improvement compared to other studies. As mentioned in the paper itself, already Bauer and Wing (2016) conducted a similar exercise (see LL 104 to 106) and the obtained results are similar. This is further highlighted by the fact that the methodology section is composed of explaining an already existing and operating model (CAPRI) and a relatively simple crop matching methodology between the estimates provided by Klein et al. (2007) and CAPRI.

The information on the agro-ecological relationship between pollination services and production is not further improved from the estimates in Klein et al. (2007), and also no detailed discussion on the possibility of establishing one is provided.

While the estimates of Klein et al. (2007) have indeed been considered state-of-the-art, the authors should make the effort to enrich these estimates with further data, which would indeed bring a new aspect to their analysis. For example, a recent paper by Siopa et al. (2024) revises the estimates of Klein and colleagues, finding that commonly applied methods for assessing pollinator dependence (PD) values can lead to underestimations. Siopa et al. also provide advice on what future studies should consider when assessing PD. This could be a good starting point for the authors of the paper at hand to introduce novel aspects into their assessment. Relying on new case studies or meta-reviews could also enrich the analysis. A novel basis for the estimation of the PD values would indeed bring new aspects into the paper, and the CAPRI model could then be used to run several scenarios based on a wider/different range of PD values, highlighting differences that emerge from different yield shock assumptions.

Nicholsen et al. (2023) could potentially also provide a good starting point for a novel approach to assess pollinators and pesticide reduction with a PE model like CAPRI. A novel idea would be, for example, establishing a link between PD estimates and pesticide reduction targets of the European Green Deal-related farm to fork and biodiversity strategies, and the resulting impact on pollinators, combined with likely yield shocks if pesticides are not applied. CAPRI could then be used to assess the market impacts of both pollinator collapse/decline and the impacts of pesticide reduction. Aizen et al. (2019) make a link between crop diversification and pollinator dependence, which also provides some interesting and novel aspects, and is a good example on how PD values (in this case also from Klein et al 2007) can be used in an analysis in combination with further aspects to provide additional information and knowledge.

In summary, the approach taken in this paper would need to be further enhanced to really provide a novelty and major breakthrough - and this could of course be done without picking up one of the specific examples mentioned above.

Additional references mentioned:

Aizen et al. (2019). Global agricultural productivity is threatened by increasing pollinator dependence without a parallel increase in crop diversification. *Global Biological Change* 25, Issue10, 3516-3527. <https://doi.org/10.1111/gcb.14736>

Nicholson, C.C., et al. (2024). Pesticide use negatively affects bumble bees across European landscapes.

Nature 628, 355–358. <https://doi.org/10.1038/s41586-023-06773-3>

Siopa, C., et al. (2024). Animal-pollinated crops and cultivars—A quantitative assessment of pollinator dependence values and evaluation of methodological approaches. *Journal of Applied Ecology*, 61, 1279–1288. <https://doi.org/10.1111/1365-2664.14634>

2. Pollinator collapse scenario

The authors need to make clear in the title and throughout the text in the paper that they model a complete pollinator collapse, not just a decline. This drastic assumption would also need to be (better) motivated and qualified (e.g. is this realistic?).

LL 119 – 122: This section 2.1.2 is confusing:

2.1 The productivity shocks inserted into CAPRI are most of the times only crop-dependant and not region-dependant, since no “new” information is used regarding the different impact of pollination on crop X in region A versus region B.

2.2 The region-specific aspect only comes into play when the crops cannot be directly matched between Klein et al (2007) and CAPRI. This needs to be clarified in the text, as the reader might be led to believe that region-specific estimates of the agro-ecological relationship between pollination services and production are used. Therefore, it should be highlighted that it is only for commodity accounts which cannot be mapped directly to the ones of Klein et al (2007) that the yield shock becomes region-specific, given the specific crop mix of a region. This needs to be made clear by first stating that for most crops it is a one-to-one match, and only for those commodities where it cannot be linked directly the following formulas, as expressed in LL 123 and LL 126, need to be used to match the crops provided by Klein et al 2007 to those of CPARI.

2.3 Ideally an example can be given (a couple of sentences) on why, for example, for “Other crops” region “Global” shows a mean of 6.2, while Europe shows 2.6.

3 Further Comments:

3.1. Likelihood of pollination collapse: Although the Discussion section alludes to the unlikelihood of a total pollination decline, the rate of decline is not discussed in detail.

3.2. Limitation of estimates provided by Klein et al 2007 need to be discussed.

3.3. Crop commodities are used along with crop and commodities along the text. Useful to remain consistent or otherwise define what the difference between crop and crop commodity is.

3.4. The abstract needs to clearly mention that it is not a worldwide decline which is analysed but instead a collapse.

3.5. To say in general terms that Easter countries are resistant to biodiversity-friendly policies is a strong statement which has to be explained in more detail in the discussion, but not mentioned in the abstract. In general, such a claim is not really assessed in your analysis, although you visually try to indicate a correlation related to the rejection of the Nature Restoration Law (NRL) proposal in the European Parliament. This is not enough, at least as long as you do not really assess the positive impacts of the NRL on pollinators. You could also make such a claim if you would show differences between specific biodiversity and pollinator-friendly policies to which certain countries are reluctant, but you do not run specific-policy scenarios on that.

3.6. Line 177ff: What are the differences between initial and final yield decline, which are attributed to intensification? How realistic are they? Why are these effects higher in the Global South? Why is the scope of cropland expansion substantially higher in regions outside of Europe?

3.7. The Discussion or Results should explain why cropland expansion happens outside of Europe.

3.8. Figure 1: given the difference in y-axis extent, the chart becomes consuming. Although it summarizes all information in one single figure, it can lead to confusion to the y-axis extent.

3.9. Global welfare loss of 302 billion EUR: how is global welfare measured in CAPRI?

3.10. L225 ff: Welfare impacts: You claim that the “consumer welfare declines overlap largely with EU Member States degree of opposition towards biodiversity conservation policies such as the Nature Restoration Law proposal”. This can be only visually inspected in the provided graphs. Please add a correlation chart in supplementary material, if such a comment is made.

3.11. You need to provide the explanation behind spatial heterogeneity on profits (LL 313-315)

3.12. The discussion mentions exacerbating food insecurity in exporting countries. Concrete examples should be given to

make this argument more plausible (LL 325). Also, explain why exporting countries have food insecurity issues. Which ones exactly and how is food security measured? It would be useful to provide statistics on food security situations when referring to a further exacerbating situation. This holds also for L296ff, i.e. which countries for example are major exporting countries to the EU and face food security challenges? Missing reference/examples and further explanations for this statement also in LL 359 – LL 360 and LL367.

3.13. The discussion lacks a more detailed comparison to other studies in the field of evaluating pollination services. LL 283 ff mention Lippert et al. and Bauer and Wing, but also other methods should be at least mentioned to put this study into context.

3.14. The structure of the sections is not aligned with those from the Nature Communication guidelines.

3.15. Figure 3: Coordinate system seems incorrect for maps.

3.16. The paper needs to undergo a language editing as it has many (although minor) spelling mistakes and some grammar issues.

(Remarks on code availability)

Reviewer #3

(Remarks to the Author)

(Remarks on code availability)

Version 1:

Reviewer comments:

Reviewer #2

(Remarks to the Author)

Dear authors,

thank you for addressing all comments. I know we were asking for rather a lot, but I think you could considerably improve the paper.

You made the effort of considering a more novel data source, do a sensitivity analysis and you simulate a scenario which as you say has not been simulated like that so far. Therefore, I think your paper has a message worth publishing.

Minor comments you should consider before publishing:

1. The model's time line is of 14 years, i.e. starting a simulation in 2016 and simulating a wild pollination collapse in 2030. When mentioning that a pollination collapse cannot be excluded, do you mean in 2030, i.e. in 5 years time or in 14 years of simulation?

2. Line 116: when using the word "strong yield gap" it would be good to make reference to the table that provides the yield gaps by crop.

3. Line 175: Clarification of food insecurity has been made, but link to FAO (reference 35) is not available).

4. Line 325: "... regions facing the greatest economic impacts from a pollinator collapse may be less supportive of biodiversity policies, potentially due to concerns about immediate economic costs."

Another reason could be the effectiveness of the mentioned biodiversity policies.

5. Discussion or Results could explain why Eastern European countries are more severely affected – i.e. ultimately, this goes back to crop mix, doesn't it?

(Remarks on code availability)

Reviewer #4

(Remarks to the Author)

General comments

The research question is really important and timely, in the sense that the data allowing to distinguish the specific impact of wild pollinators have only recently, to my knowledge, acquired a sufficient level of precision and robustness to allow undertaking such an analysis (before reading this article and the cited references, I would have even doubted that this was the case). The hypothesis that wild and managed pollinators "contribute equally to crop pollination" (line 356) may appear as a somewhat clumsy use of the article by Reilly et al (2024) whose use of the results is, as we learn in the "Methods" section (p. 20), ultimately more nuanced. I am not an ecologist, but after a quick read of this article, I note that its authors remain cautious about the robustness of their results and this "equality" estimated on a global scale obviously experiences spatial variations. On the other hand, the hypothesis (line 360-62) that it would be possible to compensate for the disappearance of wild pollinators by developing honeybee colonies does not seem to me to be consistent with the literature on the subject (see :

Garibaldi, L. A., Steffan-Dewenter, I., Winfree, R., Aizen, M. A., Bommarco, R., Cunningham, S. A., ... & Klein, A. M. (2013). Wild pollinators enhance fruit set of crops regardless of honeybee abundance. *Science*, 339(6127), 1608-1611.

Requier, F., Pérez-Méndez, N., Andersson, G. K., Blareau, E., Merle, I., & Garibaldi, L. A. (2023). Bee and non-bee pollinator importance for local food security. *Trends in ecology & evolution*, 38(2), 196-205.

Uwingabire, Z., & Gallai, N. (2024). Impacts of degraded pollination ecosystem services on global food security and nutrition. *Ecological Economics*, 217, 108068.

A formal point is that it was uncomfortable to read pages of results and discussions on a subject and with an approach, which unavoidably makes the results dependent on the method used to obtain them, before reading anything on this method. This point will probably be useless since this organization is, I believe, a request from the journal, however, it would be useful to give a few words on it at the beginning of the paper. The CAPRI model is now fairly well-known and it seems well suited to modeling the impacts of a pollinator collapse; but it was not obvious to me that it had the capacity to integrate appropriately the impact of fine differences between wild and managed pollinators, including the many substitution that will emerge in the behaviors of the final consumers.

Along the same lines, the statement (p. 5) that production declines linked to the collapse of pollinators do not translate into a decrease in producers' income by giving several figures, without explaining how they were obtained, is disturbing.

Agricultural products often benefit from the "King-Davenant law", but these effects can be erased by the existence of substitute products or in an economy open to imports.

Specific comments

P.12 lines 230-231. The sentences "... there is a noticeable overlap between the average voting behavior of parliament members of EU member states on two major biodiversity-related proposals (...) This voting behavior has a statistically significant negative correlation with the projected impact of a wild pollinator collapse on consumer welfare..." either say too much or not enough. If you want to learn something from this correlation, I think it's necessary to be more explicit about what it means to you. The observation is, moreover, interesting...

P.14 lines 264-265. The statement that a welfare reduction of 3.5% is equivalent to 0.2% GDP is somewhat surprising and could be more explained (we could expect not so different digits).

P.17 lines 362-364. The assumption of a "perfect substitutability between managed and wild pollinators" is not supported by the literature (See the references above)

P.18. The assumption of a complete wild pollinators collapse is unlikely but is not unacceptable as a research postulate to analyze the consequences. On the other hand, the hypothesis of pollination with drones remains, to my knowledge, speculation on an indefinite time horizon, contested by numerous specialists:

Potts, S. G., Neumann, P., Vaissière, B., & Vereecken, N. J. (2018). Robotic bees for crop pollination: Why drones cannot replace biodiversity. *Science of the total environment*, 642, 665-667.

Bongomin, O., Gilibrays Ocen, G., Oyondi Nganyi, E., Musinguzi, A., & Omara, T. (2020). Exponential disruptive technologies and the required skills of industry 4.0. *Journal of Engineering*, 2020(1), 4280156.

P.19 lines 427-428. Again, the statement "... with EU members states opposing pollinators-friendly policies carrying disproportionately high burden of impacts" is not precisely explain and remains somewhat disturbing.

P. 21 lines 486-487. Again, the acceptability of the technical feasibility of the substitution of wild pollinators by managed pollinators is not explained (and not widely accepted by the literature).

Line 497. It was a surprise for me to learn there is cocoa nut production in the EU (in overseas territories, I assume)

(Remarks on code availability)

Point-by-point response letter

Dear reviewers,

We greatly appreciate your thoughtful and constructive comments, which have significantly enhanced our analysis. In this revision, we have undertaken substantial efforts to address your feedback, and we believe this has resulted in a much-improved manuscript. The key changes are summarized as follows:

- 1. Focus on a European wild pollinator collapse:** In response to Reviewer 1, we have revised the scope from a global pollinator collapse to a large-scale collapse of wild pollinators within Europe. This analysis utilizes the latest crop-specific pollinator dependence ratios, as suggested by Reviewers 2 and 3, and the relative contribution of wild pollinators. Additionally, we contrast our results with the dependence ratios derived from the previously used dataset for this type of analysis. To our knowledge, this is the first study to assess such a scenario at this spatial scale using a detailed, global agri-food simulation model.
- 2. Critical evaluation of assumptions and sensitivity analysis:** We now include a 95% confidence interval for the magnitude of the wild pollinator collapse and conduct a systematic sensitivity analysis of key model parameters to assess the robustness of our results. In the discussion, we critically contrast the collapse scenario by 2030 with existing evidence on wild pollinator population trends in Europe. This includes a review of the available literature, presented in the Supplementary Methods. Furthermore, we compare our model-based results with alternative approaches, including replacement cost estimates and willingness-to-pay assessments.
- 3. Strengthened policy implications:** We have expanded our discussion of policy implications by linking our findings to existing agri-environmental measures under the Common Agricultural Policy (CAP) in Europe. Notably, we highlight that countries opposing EU pollinator-friendly policy proposals (e.g., the Nature Restoration Law and Sustainable Use Regulation) are projected to experience disproportionately high welfare losses. We identify this as a critical observation warranting future political-economic research.

In the following, you find our point-by-point responses to the specific comments.

We sincerely thank you for your time to review this paper and look forward to your comments and suggestions.

	Reviewer comments	Authors' response
	Reviewer #1	
1/1	Thank you for the invitation to review this article. In this paper, the authors model a total global collapse of both managed and wild pollinators on food production, land use and nutrient consumption. They assume no possible substitution to other forms of pollination (hand or mechanical) and take the global pollination dependency ratio as fixed, based on Klein et al (2007). The innovation of this paper is that they use a detailed partial equilibrium global agricultural model (CAPRI) to simulate the effect of this shock on food prices and consumption, allowing for changes in intensification, cropped area, international trade and crop substitution to mitigate the production shock. Because their model allows for these mitigating effects, their results show smaller economic costs than earlier work. Even so, their results highlight the wide-ranging effects of such a loss in pollinators, inducing a 6.1% increase in land cultivation, a shift to grains and meats and a 0.4% decrease in global GDP.	We thank the reviewer for their helpful comments. Based on this report and by the comments from reviewer #2 and #3, we decided to substantially revise the paper. We now model a collapse in wild pollinators restrained to Europe, as suggested by you, but which also suits the model's focus on the EU agricultural sector. The revised version now uses most recent data on dependence ratios from Siopa et al. (2024)¹ https://doi.org/10.1111/1365-2664.14634, and identifies the contribution of wild pollinators using data from Reilly et al (2024)²- https://onlinelibrary.wiley.com/doi/10.1111/geb.13843
1/2	The benefit of the paper is that by using a global model that has relatively detailed agricultural product classifications, they are better able to capture the detailed pollination dependence by crop. That said, given that pollination dependent crops often have relatively non-pollination dependent alternatives, their still relatively broad crop classifications will miss out on some likely substitutions (pistachios for almonds for example).	The CAPRI model covers about 50 primary and secondary agricultural commodities of which 13 commodities are pollination dependent. Our scenario shocks ten of these 14 commodities, because oilpalm, coffee and cocoa are not cultivated in Europe. The observation regarding substitution between affected and non-affected crops is well taken. However, the model's commodities that represent groups of crops (e.g. Other Vegetables – or Other Fruits and Nuts) cover a wide range of crops including both pollination dependent and independent ones. Within this group the model considers a perfect substitution, and an imperfect substitution is modelled if crops are in separate commodities (e.g. rapeseed and sunflower). So, the provided example that the model misses “out on some likely substitutions (pistachios for almonds for example)” is not true, but rather the exact opposite. This leads rather to an overestimation of the ease to which demand will react to changes in prices.
1/3	They acknowledge that the assumption that both wild and managed pollinators will be wiped out is quite strong. I appreciate that they do not have data to model the decline of only wild pollinators except for some crops, but this is a large enough assumption that it might warrant some alternatives as robustness tests. For example, instead of productivity loss, they might consider using the additional costs of increasing the number of managed pollinators as a lower bound.	Thank you for this comment which inspired us to revise the scenario. We now use data on wild pollinators and only shock them. Also, our discussion section now contrasts the welfare results with the possible replacement costs arising by increasing managed pollinators – see section 3.2.
1/4	One aspect of their model that was not perfectly clear was that farmers are allowed to mitigate the loss in productivity from pollination by intensifying production. As they note, this mechanism is particularly important for sub-	Since in the revised version we do only simulate a productivity shock on European countries, we do not anymore observe the previous results in the Global South, particularly in sub-Saharan Africa.

	Saharan Africa. What was not clear was the assumptions made in the model about the production function and how other inputs interact with the loss of pollination services. This relationship might be quite different for different crops.	We have now included a better description of the CAPRI model’s endogenous yield mechanisms, see also Supplementary Table 2, which also shows – as you have suggested – that the endogenous yield changes differ across crops. We also describe this in the main text, see L. 97ff: “The price increases also lead to an endogenous change in crop yields and incentivizes land reallocation from low-input to high-input technologies. The combined effect of these model mechanisms buffers the simulated production shocks for pollination dependent crops by about 2.1% on average across all European regions (Supplementary Table 2).”
1/5	By construction, the paper ignores other ecological costs of pollinator loss such as their effect on non-cultivated plants and as part of a food chain. Further, earlier work shows that natural areas serve as wild pollinator habitat, so that farmland expansion from a reduction in pollinators might trigger a negative feedback loop. Similar threats to pollinators may arise from intensification itself. Assuming a partial pollinator decline, some discussion of these other effects might be useful (e.g. Potts et al 2016).	We thank you for this comment and we fully agree. Our revised discussion (L. 408ff) now includes the following paragraph including the Potts et al.³ reference: “Finally, the ecological and social costs of a wild pollinator collapse extend beyond agriculture, potentially disrupting ecological processes, food webs, and overall biodiversity resilience^{3,4}. Wild pollinators are essential for the reproduction of most wild plant species and their diversity⁵⁻⁷. These wild plants provide food for granivorous species, while wild pollinators themselves serve as a food source for higher trophic levels^{8,9}. Hence, although the agricultural impacts can be monetized, the broader, interconnected ecological losses remain invaluable for long-term stability.” We also mention this aspect once again in our concluding remarks. The aspect of negative feedback loops is now addressed at L. 300: “Globally, and particularly within Europe, the model projects cropland expansion that increases pressures on biodiversity^{10,11} and potentially triggering negative feedback loops that affect other ecosystem services, such as natural pest control¹²”
1/6	Last, I struggled a bit in terms of the story the authors are trying to tell. I appreciated the authors’ motivation based on specific legislative attempts to preserve biodiversity and wild pollinators in Europe, and the findings that some of the locations most severely hurt by pollinator loss are those areas that do not support these interventions. The simulation they run, however, is a global collapse of pollinators, where the link to EU specific legislation is less clear. They might consider instead a narrower simulation of the effect of a EU pollinator collapse on global agricultural production and consumption. Alternatively, they might consider how a global collapse might be mediated by pollinator preservation in the	Thank you for your comments and suggestions. We gave them considerable thoughts, and this is why we eventually decided to focus on a wild pollinator collapse that is restrained to the European continent and not to the whole world. This is of course still a hypothetical scenario, potentially reflecting a case in which all other regions manage to protect pollinators except for Europe. In the discussion, we now also contrast the welfare results of this scenario to both replacement costs through managed pollinators and the cost of pollinator-friendly agri-environmental measures at similar levels of the simulated welfare losses.

	EU. Or they might cite efforts in other locations to protect pollinator habitat to look at global costs and benefits of such efforts.	However, we consider it beyond the scope of the paper to assess the conservation effects of such measures.
1/7	In summary, this paper contributes to our understanding of the importance of pollinators to the global agri-food system by modeling the effect of a production shock to pollination-dependent crops, allowing for farmers and consumers to respond by changing crops, production practices, land area and consumption. They also highlight the regional distribution of the impact of these changes. The drawback of the article is that the authors assume a complete loss of pollination, no change in crop breeding or other measures that could reduce pollination dependence, and by the nature of a large global model of agricultural production, still have relatively broad crop categories that may mask further substitution effects. I am uncertain whether this contribution is sufficient for Nature Communications, but I do believe that the paper makes valuable contributions to the literature on pollinator impact. Potts, S., Imperatriz-Fonseca, V., Ngo, H. et al. Safeguarding pollinators and their values to human well-being. Nature 540, 220–229 (2016). https://doi-org.proxy.library.ucsb.edu/10.1038/nature20588	We thank you for your review and the helpful comments. We hope that our revision of the paper addresses the drawbacks of our earlier paper and that it meets your expectations. Also, we would once more highlight that the model’s crop disaggregation is detailed, especially in contrast to the only (somewhat) comparable study by Bauer and Wing (2016)¹³ (10.1016/j.ecolecon.2016.01.011) which simulated a global collapse of wild and managed pollinators. The model in Bauer and Wing had only four pollination dependent commodities, while the CAPRI model has 13 as explained above. In comparison, the CAPRI model also has a far more detailed coverage of non-pollination dependent crops. Also, as mentioned above, we argue that by grouping commodities we overestimate the substitution ability of the agri-food system, which means the real effects on would rather be more adverse being an argument that our analysis – in this respect – is rather conservative.

Reviewer #2		
2/1	The paper “Impacts of Pollinator Declines on Agri-Food Systems in Europe and Beyond”, assesses the possible effects on agri-food markets of a worldwide collapse of pollinators by 2030. For the assessment, the authors rely on estimates on pollination dependency ratios from Klein et al. (2007), and they apply respective yield shocks in the CAPRI model.	Thank you for the careful reading of our paper and for providing many helpful comments and suggestions. Please note, that given these comments we have decided to considerably revise the paper – see our next comment.
Major Comments		
2/2	1. Novelty and Transformation: In general, the approach of using pollinator-related yield shocks of another study into the partial equilibrium model CAPRI is sound. However, the major concern with the presented work is that, in my opinion, it lacks novelty and a major breakthrough as it uses estimates on pollination dependency from a study in 2007 and inserts them into an existing partial equilibrium model without any model improvements. The authors fail to highlight in the text why this work provides a major improvement compared to other studies. As mentioned in the paper itself, already Bauer and Wing (2016) conducted a similar exercise (see LL 104 to 106) and the obtained results are similar. This is further highlighted by the fact that the methodology section is composed of explaining an already existing and operating model (CAPRI) and a relatively simple crop matching methodology between the estimates provided by Klein et al. (2007) and CAPRI. The information on the agro-ecological relationship between pollination services and production is not further improved from the estimates in Klein et al. (2007), and also no detailed discussion on the possibility of establishing one is provided. While the estimates of Klein et al. (2007) have indeed been considered state-of-the-art, the authors should make the effort to enrich these estimates with further data, which would indeed bring a new aspect to their analysis. For example, a recent paper by Siopa et al. (2024) revises the estimates of Klein and colleagues, finding that commonly applied methods for assessing pollinator dependence (PD) values can lead to underestimations. Siopa et al. also provide advice on what future studies should consider when assessing PD. This could be a good starting point for the authors of the paper at hand to introduce novel aspects into their assessment. Relying on new case studies or meta-reviews could also enrich the analysis. A novel basis for the estimation of the PD values would indeed bring new aspects into the paper, and the CAPRI model could then be used to run several scenarios based on a wider/different range of PD values, highlighting differences that emerge from different yield shock assumptions.	Many thanks for this extensive comment. We agree with your point that the Klein et al (2007)¹⁴ data is outdated and rather rough with respect to the actual degree of dependence. We absolutely agree with your suggestion and have now used the Siopa et al (2024)¹ data. Being inspired by your suggestion to look for further meta- analysis we now only consider the fraction of pollination services provided by wild pollinators using the data from Reilly et al. (2024)². Based on this, we now simulate a collapse of wild pollinators, which we constrain to the European continent following the advice from reviewer #1. This has the benefit that we better leverage the comparative strength of the CAPRI model, which is focused on the European agricultural sector and activities Europe embedded in the global agri-food system. Following your suggestion, in addition to our “main scenario” that uses the Siopa et al. data we also run an alternative scenario based on the older data on dependence ratios from Klein et al. (2007). Supplementary Table 3 includes a detailed comparison of the welfare results when using either of these two datasets on dependence ratios. We also briefly discuss this finding in the main text at L. 293ff: “Simulating the same scenario with the alternative, but older data on dependence ratios from Klein et al.¹⁴ would result in 20.6% lower global economic welfare losses of €27.3 billion (Supplementary Table 2) reflecting the mostly lower dependence ratios of this data compared to the most recent data provided by Siopa et al.¹ (see difference in productivity shocks in Supplementary Table 1).” We have refrained from extending the CAPRI model, e.g., to capture agro-ecological relationships, both because of lack of data (see below) but also because this was not the initial objective of this study.

Nicholsen et al. (2023) could potentially also provide a good starting point for a novel approach to assess pollinators and pesticide reduction with a PE model like CAPRI. A novel idea would be, for example, establishing a link between PD estimates and pesticide reduction targets of the European Green Deal-related farm to fork and biodiversity strategies, and the resulting impact on pollinators, combined with likely yield shocks if pesticides are not applied. CAPRI could then be used to assess the market impacts of both pollinator collapse/decline and the impacts of pesticide reduction. Aizen et al. (2019) make a link between crop diversification and pollinator dependence, which also provides some interesting and novel aspects, and is a good example on how PD values (in this case also from Klein et al 2007) can be used in an analysis in combination with further aspects to provide additional information and knowledge.

In summary, the approach taken in this paper would need to be further enhanced to really provide a novelty and major breakthrough - and this could of course be done without picking up one of the specific examples mentioned above.

Additional references mentioned:

Aizen et al. (2019). Global agricultural productivity is threatened by increasing pollinator dependence without a parallel increase in crop diversification. *Global Biological Change* 25, Issue10, 3516-3527. <https://doi.org/10.1111/gcb.14736>

Nicholson, C.C., et al. (2024). Pesticide use negatively affects bumble bees across European landscapes.

Nature 628, 355–358. <https://doi.org/10.1038/s41586-023-06773-3>

Siopa, C., et al. (2024). Animal-pollinated crops and cultivars—A quantitative assessment of pollinator dependence values and evaluation of methodological approaches. *Journal of Applied Ecology*, 61, 1279–1288. <https://doi.org/10.1111/1365-2664.14634>

We have also considered your suggestions regarding incorporating linkages between changes in pesticide use and supply of pollination services. However, the problems here are manifold. First, the CAPRI model only knows one pesticide type – so a policy scenario would need to focus on reducing all types of pesticides equally. Much more important, the literature on pesticide impacts on pollination supply is still evolving and not very conclusive. For instance, the Environmental Risk Assessment (ERA) scheme for registration of pesticides in the European Union in most cases is not addressing effects on food-webs and ecosystem (Brühl and Zaller 2019,¹⁵ <https://doi.org/10.3389/fenvs.2019.00177>). There are many studies on the impact of pesticides on managed and wild-pollinators (e.g., Battisti et al.¹⁶ <https://doi.org/10.1016/j.scitotenv.2021.145397> or Tosi et al.¹⁷ [10.1016/j.scitotenv.2022.156857](https://doi.org/10.1016/j.scitotenv.2022.156857)), but only few studies on the pesticide effect of specific pesticides on the provision of pollination services to specific crops (see e.g., Stanley et al. 2015¹⁸ for a study on neonicotinoids on bumble bees and pollination services to apples - <https://doi.org/10.1038/nature16167>). Nicholsen et al. (2023)¹⁹ focus on bumble bees and report colony health changes as one main result, but we find it very difficult if not impossible to generalize this finding.

The literature is expanding in this direction, see also a recent review on the pesticide exposure and effects on non-apids bees (Raine & Rundlöf 2024²⁰, <https://doi.org/10.1146/annurev-ento-040323-020625>). However, we would require synthesized information about how different pesticides (applied in different crops) impact wild pollinators in general across a broad set of main commercial agricultural crops. To the best of our knowledge, this information is not yet available.

According to our reading, Aizen et al. (2019)²¹ investigate the relationship between changes in a countries' reliance on pollination services and changes in agricultural practices and crop diversity. However, the Aizen et al. study does not provide information on the relationship between impacts of preserving crop diversity on pollinator populations.

We discuss the findings of Aizen et al., which in some respect are similar to our analysis of EU member states voting behavior. In line L.. we write:

The significant correlation between consumer surplus losses and EU member states' resistance to pollinator-friendly policies, measured by the average voting behavior of EU parliament members, is an incidental finding that warrants further investigation. This relationship suggests that regions facing the greatest

		economic impacts from a pollinator collapse may be less supportive of biodiversity policies, potentially due to concerns about immediate economic costs. A similar pattern was observed in a study that showed that countries in the Americas and Asia, despite their increasing reliance on pollination-dependent crops, often expand agricultural practices that undermine pollination services²¹. We also highlight the role of crop diversity being an important factor for pollinator conservation and highlight it in the revised discussion, see L. 339 ff.
2/3	2. Pollinator collapse scenario The authors need to make clear in the title and throughout the text in the paper that they model a complete pollinator collapse, not just a decline. This drastic assumption would also need to be (better) motivated and qualified (e.g. is this realistic?).	We have made sure that we consistently refer to a “collapse” and not a “decline” which would indeed be misleading. We qualify the choice of the extreme scenario in the “limitations” section starting at L.378 where we write: “Several limitations affect our model and simulations. First, we simulate a complete collapse of wild pollinators, representing a hypothetical and extreme scenario, despite mounting evidence of significant insect declines in Europe and globally^{22–24}. We define a wild pollinator collapse as a 90% decline in population over the model’s time horizon (14 years), corresponding to an annual decline rate of 15.2%. There is a notable lack of comprehensive data on pollinator population trends in Europe. The available evidence suggests annual rates of decline ranging from 0.8% to 14.1%, with an average of 5.8% (Supplementary Methods). These estimates vary widely in terms of taxa, geographic coverage, and methodology. They include studies on hoverfly (Syrphidae) abundance^{22,25} and bee and hoverfly occupancy^{26,27} – both key pollinator taxa²⁸ – as well as studies on trends in the abundance or biomass of broader groups such as flying insects^{29,30}, arthropods²⁴, and terrestrial insects³¹. Extrapolating these trends to 2030 in absence of mitigation efforts indicates that a wild pollinator collapse in Europe, as defined in our study, is unlikely but cannot be entirely ruled out (Supplementary Methods). Although extreme, investigating this “what if” scenario provides valuable insights into how severe pollination service declines might impact regions, markets, and stakeholders, informing biodiversity conservation strategies, land-use planning, and agricultural policy. Additionally, it highlights the urgent need for spatially comprehensive, long-term monitoring of wild pollinators in Europe to better assess the plausibility of such extreme scenarios.”

2/4	LL 119 – 122: This section 2.1.2 is confusing: 2.1 The productivity shocks inserted into CAPRI are most of the times only crop-dependant and not region-dependant, since no “new” information is used regarding the different impact of pollination on crop X in region A versus region B. 2.2 The region-specific aspect only comes into play when the crops cannot be directly matched between Klein et al (2007) and CAPRI. This needs to be clarified in the text, as the reader might be led to believe that region-specific estimates of the agro-ecological relationship between pollination services and production are used. Therefore, it should be highlighted that it is only for commodity accounts which cannot be mapped directly to the ones of Klein et al (2007) that the yield shock becomes region-specific, given the specific crop mix of a region. This needs to be made clear by first stating that for most crops it is a one-to-one match, and only for those commodities where it cannot be linked directly the following formulas, as expressed in LL 123 and LL 126, need to be used to match the crops provided by Klein et al 2007 to those of CPARI.	We agree that our text was confusing here. The documentation of how the productivity shock is derived was completely revised and we now take pain to make clear when a shock is just commodity specific and when it is also country-specific. We now write in section 4.1.2 (L. 509ff) “The productivity shocks are derived for each of the ten pollination-dependent commodities that are included in CAPRI and produced in Europe. For those commodities that do not refer to a single but multiple crops (e.g., commodities like “Other Vegetables” or “Other Fruits”), the productivity shock is determined by the region-specific production value shares of the relevant pollination dependent crops in the respective commodity. The productivity shock, $\delta_{c,n}$, for a commodity c and country (or region) n, is calculated as: $\delta_{c,n} = 1 - \sum_i^I shrcom_{c,i,n} \varphi_c D_i$ where φ_c is the relative contribution of wild pollinators, D_i is the overall dependence ratio on pollination services by both managed and wild pollinations, and $shrcom_{c,i,n}$ is the share of crop i in total production value of a CAPRI commodity c in country n. The shocks are identical across countries if the CAPRI agricultural commodity is directly mapped to a single crop included in the Siopa et al. data. This applies for four commodities: Rapeseed, Sunflower, Soya and Tomatoes. For the remaining six pollination-dependent commodities, the shocks are country-specific because multiple crops are aggregated in one commodity. The aggregation shares rely on information on production value taken from the FAOSTAT database³² for the model’s base year 2017. The calculation of the productivity shocks was performed using the GAMS software, for which the code is included in the electronic supplementary materials.”
2/5	2.3 Ideally an example can be given (a couple of sentences) on why, for example, for “Other crops” region “Global” shows a mean of 6.2, while Europe shows 2.6.	We do not report this result (or any similar one) anymore since we have substantially changed the model scenario. See comments above.

2/6	3 Further Comments: 3.1. Likelihood of pollination collapse: Although the Discussion section alludes to the unlikelihood of a total pollination decline, the rate of decline is not discussed in detail.	We agree that this was a shortcoming of our paper and have put efforts in contextualizing our scenario and we now extensively contrast the choice of our scenario with the available evidence. Please refer to our response made to your comment 2/3 where we already cited the relevant passage of the revised text. The Supplementary Methods contain more details.
-----	--	--

2/7	3.2. Limitation of estimates provided by Klein et al 2007 need to be discussed.	We address the limitations inherent in dependence ratios in our revised discussion, where we write in L. 400: “Our estimates for crop dependence on pollination vary based on conservative assumptions, modeling the lower and upper bounds for productivity shocks and conducting sensitivity analyses. Dependence ratios come from studies with differing methodologies (e.g., variations in the quality of “open pollination” references) and often overlook variations in crop quality and variety^{1,33,34}. This study conservatively excludes potential crop quality deterioration and wild pollinator contributions to seed production, a factor crucial for forage crops and vegetables that accounts for up to 25% of total pollination service value.^{35,36}”
2/8	3.3. Crop commodities are used along with crop and commodities along the text. Useful to remain consistent or otherwise define what the difference between crop and crop commodity is.	Thank you. This has been changed. We do not refer to “crop commodity” anymore.
2/9	3.4. The abstract needs to clearly mention that it is not a worldwide decline which is analysed but instead a collapse.	This has been done.
2/10	3.5. To say in general terms that Eastern countries are resistant to biodiversity-friendly policies is a strong statement which has to be explained in more detail in the discussion, but not mentioned in the abstract. In general, such a claim is not really assessed in your analysis, although you visually try to indicate a correlation related to the rejection of the Nature Restoration Law (NRL) proposal in the European Parliament. This is not enough, at least as long as you do not really assess the positive impacts of the NRL on pollinators. You could also make such a claim if you would show differences between specific biodiversity and pollinator-friendly policies to which certain countries are reluctant, but you do not run specific-policy scenarios on that.	We have substantially revised this section. We now look at the correlation between welfare results in EU member states and their voting behavior regarding two important pollinator/biodiversity conservation policies (NRL and the Sustainable Use Regulation, SUR). We show that there is a negative and statistically significant correlation. We highlight this as an interesting pattern and discuss the possible meaning of it. Since our scenario is counterfactual, we can only speculate why we see this correlation. In this context, we do not explicitly mention Eastern countries anymore. The revised abstract still mentions this finding, where we now write: “Global welfare losses would reach €34 billion, with Europe and the EU accounting for €24 billion and €12 billion, disproportionately impacting EU member-states resistant to biodiversity-friendly policies.” We think it is an interesting observation, that countries with higher resistance towards biodiversity friendly policies would be disproportionately affected by a hypothetical collapse of wild pollinators. We think it is legitimate to highlight this observation, and we now show that this is not only a visual impression, but it is also statistically significant for the main two policy proposals. However, if this argument is deemed insufficiently compelling, we are open to excluding this finding from the abstract.

		Modeling the possible (positive) impacts of the NRL (or now also the SUR) would indeed be very interesting. However, we think that this should better be addressed by studies that pick-up on our incidental finding on the linkage between welfare results and voting behavior – as this was not the objective of this study.
2/11	3.6. Line 177ff: What are the differences between initial and final yield decline, which are attributed to intensification? How realistic are they? Why are these effects higher in the Global South? Why is the scope of cropland expansion substantially higher in regions outside of Europe?	The differences between initial and final yield decline is due to the endogenous changes in CAPRI for which we now added more information in Supplementary Table 2. In the main text, we write now (L. 97ff): “The price increases also lead to an endogenous change in crop yields and incentivizes land reallocation from low-input to high-input technologies. The combined effect of these model mechanisms buffers the simulated production shocks for pollination dependent crops by about 2.1% on average across all European regions (Supplementary Table 2).” The cropland expansion was higher outside of Europe because the model assumes much higher land supply elasticities for most regions in the Global South, where conversion of non-agricultural land (mostly forests) is mostly taking place. We could explain this in more detail, but this effect is not as relevant anymore given the change in the scenario.
2/12	3.7. The Discussion or Results should explain why cropland expansion happens outside of Europe.	With our revised analysis substantially less cropland expansion is taking place, simply because a collapse in (only) wild pollinators restrained in Europe is much less drastic compared to a global collapse in all pollinators. But the reason for more elastic cropland expansion outside of Europe is because of the higher land supply elasticities (see point above), which again is also e.g., because governance of land tenure and measures against deforestation are less strictly enforced in lower-income regions.
2/13	3.8. Figure 1: given the difference in y-axis extent, the chart becomes consuming. Although it summarizes all information in one single figure, it can lead to confusion to the y-axis extent.	We agree that the different y-axis scales can be confusing. We tested a version in which all facets have the same y-axis extent, but this would make the graph unreadable. We added a note to Fig. 1 to make the reader aware that the y-axis scales are specific to each indicator.
2/14	3.9. Global welfare loss of 302 billion EUR: how is global welfare measured in CAPRI?	We have extended the methods section to explain how total welfare is composed of and how the most relevant components (consumer and producer surplus) are measured. For more details, we refer to the extensive CAPRI model documentation.
2/15	3.10. L225 ff: Welfare impacts: You claim that the “consumer welfare declines overlap largely with EU Member States degree of opposition towards biodiversity conservation policies such as the Nature Restoration Law proposal”. This can be	We have added a correlation chart between welfare changes and the average voting behavior of EU parliament members (grouped by member state) on the Nature Restoration Law and the Sustainable Use Regulation (See Fig. 6). We also

	only visually inspected in the provided graphs. Please add a correlation chart in supplementary material, if such a comment is made.	highlight that this was rather an incidental finding after analyzing the spatial distribution of the welfare effects. We discuss the need for further investigation in the discussion and acknowledge that more thorough analysis is needed.
2/16	3.11. You need to provide the explanation behind spatial heterogeneity on profits (LL 313-315)	This finding is not included any more due to the overall revision of the paper.
2/17	3.12. The discussion mentions exacerbating food insecurity in exporting countries. Concrete examples should be given to make this argument more plausible (LL 325). Also, explain why exporting countries have food insecurity issues. Which ones exactly and how is food security measured? It would be useful to provide statistics on food security situations when referring to a further exacerbating situation. This holds also for L296ff, i.e. which countries for example are major exporting countries to the EU and face food security challenges? Missing reference/examples and further explanations for this statement also in LL 359 – LL 360 and LL367.	The impact on food security in countries outside of Europe are much lower in the revised scenario. Nevertheless, we show that through the higher exports to Europe, despite the simultaneous increase in production, there is less nutrient-dense food available in the exporting countries. In addition, we cite the latest FAO report on food security which clearly shows that the regions from which these exports are sourced from suffer from food insecurity. Here we write in L.179ff: “Globally, while most food-insecure people live outside Europe, the decline in Europe’s pollinator-dependent crop production and its shift in trade have negative spillover effects. Although regions outside Europe might see a production increase in micronutrient-rich crops, the intensified import demand from Europe would more than offset these gains, leading to a net decline in nutrient-dense food consumption worldwide (Fig. 2). This reduction would exacerbate food insecurity in regions already vulnerable to malnutrition (Supplementary Figure 3), particularly in Africa, as well as parts of Middle and South America and Asia with high rates of food insecurity³⁷.” Of course, the exact impact on food security is difficult to assess because the model only considers one household type (which we also acknowledge) and therefore we cannot clearly assess how higher food prices and lower food availability would impact food insecure households at the margin. Also, those households that are food producers would benefit because of higher food prices – see also Supplementary Figure 1.
2/18	3.13. The discussion lacks a more detailed comparison to other studies in the field of evaluating pollination services. LL 283 ff mention Lippert et al. and Bauer and Wing, but also other methods should be at least mentioned to put this study into context.	We agree and we have revised the discussion and we now cite studies using other methods, incl. beehive rent, replacement cost and stated preference. See also our response to 2/7. Also, we cite the review of Breeze et al. 2016.³⁸ https://doi.org/10.1016/j.tree.2016.09.002
2/19	3.14. The structure of the sections is not aligned with those from the Nature Communication guidelines.	We have checked the “author guide” and should now comply with the guidelines.
2/20	3.15. Figure 3: Coordinate system seems incorrect for maps.	Thank you this has been corrected.

2/21	3.16. The paper needs to undergo a language editing as it has many (although minor) spelling mistakes and some grammar issues.	Our manuscript has been thoroughly proofread and copy-edited by a native speaker, but any remaining errors or issues are, of course, our own.
Reviewer #3 (Remarks to the Author):		
3/1	I co-reviewed this manuscript with one of the reviewers who provided the listed reports. This is part of the Nature Communications initiative to facilitate training in peer review and to provide appropriate recognition for Early Career Researchers who co-review manuscripts.	We also thank you for your comments and suggestions. We hope we have adequately addressed them.

References used in the responses:

1. Siopa, C., Carvalho, L. G., Castro, H., Loureiro, J. & Castro, S. Animal-pollinated crops and cultivars—A quantitative assessment of pollinator dependence values and evaluation of methodological approaches. *J. Appl. Ecol.* **61**, 1279–1288 (2024) [https://doi.org/ 10.1111/1365-2664.14634](https://doi.org/10.1111/1365-2664.14634).
2. Reilly, J. R. *et al.* Wild insects and honey bees are equally important to crop yields in a global analysis. *Glob. Ecol. Biogeogr.* **33**, e13843 (2024) [https://doi.org/ 10.1111/geb.13843](https://doi.org/10.1111/geb.13843).
3. Potts, S. G. *et al.* Safeguarding pollinators and their values to human well-being. *Nature* **540**, 220–229 (2016) [https://doi.org/ 10.1038/nature20588](https://doi.org/10.1038/nature20588).
4. Reilly, J. R. *et al.* Crop production in the USA is frequently limited by a lack of pollinators. *Proc. R. Soc. B Biol. Sci.* **287**, 20200922 (2020) [https://doi.org/ 10.1098/rspb.2020.0922](https://doi.org/10.1098/rspb.2020.0922).
5. Ollerton, J., Winfree, R. & Tarrant, S. How many flowering plants are pollinated by animals? *Oikos* **120**, 321–326 (2011) [https://doi.org/ 10.1111/j.1600-0706.2010.18644.x](https://doi.org/10.1111/j.1600-0706.2010.18644.x).
6. Rodger, J. G. *et al.* Widespread vulnerability of flowering plant seed production to pollinator declines. *Sci. Adv.* **7**, eabd3524 (2021) [https://doi.org/ 10.1126/sciadv.abd3524](https://doi.org/10.1126/sciadv.abd3524).
7. Wei, N. *et al.* Pollinators contribute to the maintenance of flowering plant diversity. *Nature* **597**, 688–692 (2021) [https://doi.org/ 10.1038/s41586-021-03890-9](https://doi.org/10.1038/s41586-021-03890-9).
8. Bowler, D. E., Heldbjerg, H., Fox, A. D., de Jong, M. & Böhning-Gaese, K. Long-term declines of European insectivorous bird populations and potential causes. *Conserv. Biol. J. Soc. Conserv. Biol.* **33**, 1120–1130 (2019) [https://doi.org/ 10.1111/cobi.13307](https://doi.org/10.1111/cobi.13307).
9. Tallamy, D. W. & Shriver, W. G. Are declines in insects and insectivorous birds related? *Ornithol. Appl.* **123**, duaa059 (2021) [https://doi.org/ 10.1093/ornithapp/duaa059](https://doi.org/10.1093/ornithapp/duaa059).
10. Millard, J. *et al.* Key tropical crops at risk from pollinator loss due to climate change and land use. *Sci. Adv.* **9**, eadh0756 (2023) [https://doi.org/ 10.1126/sciadv.adh0756](https://doi.org/10.1126/sciadv.adh0756).

11. Dicks, L. V. *et al.* A global-scale expert assessment of drivers and risks associated with pollinator decline. *Nat. Ecol. Evol.* **5**, 1453–1461 (2021) <https://doi.org/10.1038/s41559-021-01534-9>.
12. Klinnert, A. *et al.* Landscape features support natural pest control and farm income when pesticide application is reduced. *Nat. Commun.* **15**, 5384 (2024) <https://doi.org/10.1038/s41467-024-48311-3>.
13. Bauer, D. M. & Sue Wing, I. The macroeconomic cost of catastrophic pollinator declines. *Ecol. Econ.* **126**, 1–13 (2016) <https://doi.org/10.1016/j.ecolecon.2016.01.011>.
14. Klein, A.-M. *et al.* Importance of pollinators in changing landscapes for world crops. *Proc. Biol. Sci.* **274**, 303–313 (2007) <https://doi.org/10.1098/rspb.2006.3721>.
15. Brühl, C. A. & Zaller, J. G. Biodiversity Decline as a Consequence of an Inappropriate Environmental Risk Assessment of Pesticides. *Front. Environ. Sci.* **7**, 177 (2019) <https://doi.org/10.3389/fenvs.2019.00177>.
16. Battisti, L. *et al.* Is glyphosate toxic to bees? A meta-analytical review. *Sci. Total Environ.* **767**, 145397 (2021) <https://doi.org/10.1016/j.scitotenv.2021.145397>.
17. Tosi, S., Sfeir, C., Carnesecchi, E., vanEngelsdorp, D. & Chauzat, M.-P. Lethal, sublethal, and combined effects of pesticides on bees: A meta-analysis and new risk assessment tools. *Sci. Total Environ.* **844**, 156857 (2022) <https://doi.org/10.1016/j.scitotenv.2022.156857>.
18. Stanley, D. A. *et al.* Neonicotinoid pesticide exposure impairs crop pollination services provided by bumblebees. *Nature* **528**, 548–550 (2015) <https://doi.org/10.1038/nature16167>.
19. Nicholson, C. C. *et al.* Pesticide use negatively affects bumble bees across European landscapes. *Nature* **628**, 355–358 (2024) <https://doi.org/10.1038/s41586-023-06773-3>.
20. Raine, N. E. & Rundlöf, M. Pesticide Exposure and Effects on Non-*Apis* Bees. *Annu. Rev. Entomol.* **69**, 551–576 (2024) <https://doi.org/10.1146/annurev-ento-040323-020625>.
21. Aizen, M. A. *et al.* Global agricultural productivity is threatened by increasing pollinator dependence without a parallel increase in crop diversification. *Glob. Change Biol.* **25**, 3516–3527 (2019) <https://doi.org/10.1111/gcb.14736>.
22. Hallmann, C. A., Ssymank, A., Sorg, M., de Kroon, H. & Jongejans, E. Insect biomass decline scaled to species diversity: General patterns derived from a hoverfly community. *Proc. Natl. Acad. Sci.* **118**, e2002554117 (2021) <https://doi.org/10.1073/pnas.2002554117>.

23. Müller, J. *et al.* Weather explains the decline and rise of insect biomass over 34 years. *Nature* 1–6 (2023) doi:10.1038/s41586-023-06402-z [https://doi.org/ 10.1038/s41586-023-06402-z](https://doi.org/10.1038/s41586-023-06402-z).
24. Seibold, S. *et al.* Arthropod decline in grasslands and forests is associated with landscape-level drivers. *Nature* **574**, 671–674 (2019) [https://doi.org/ 10.1038/s41586-019-1684-3](https://doi.org/10.1038/s41586-019-1684-3).
25. Barendregt, A., Zeegers, T., Van Steenis, W. & Jongejans, E. Forest hoverfly community collapse: Abundance and species richness drop over four decades. *Insect Conserv. Divers.* **15**, 510–521 (2022) [https://doi.org/ 10.1111/icad.12577](https://doi.org/10.1111/icad.12577).
26. Powney, G. D. *et al.* Widespread losses of pollinating insects in Britain. *Nat. Commun.* **10**, 1018 (2019) [https://doi.org/ 10.1038/s41467-019-08974-9](https://doi.org/10.1038/s41467-019-08974-9).
27. Cooke, R. *et al.* Protected areas support more species than unprotected areas in Great Britain, but lose them equally rapidly. *Biol. Conserv.* **278**, 109884 (2023) [https://doi.org/ 10.1016/j.biocon.2022.109884](https://doi.org/10.1016/j.biocon.2022.109884).
28. Rader, R. *et al.* Non-bee insects are important contributors to global crop pollination. *Proc. Natl. Acad. Sci.* **113**, 146–151 (2016) [https://doi.org/ 10.1073/pnas.1517092112](https://doi.org/10.1073/pnas.1517092112).
29. Hallmann, C. A. *et al.* More than 75 percent decline over 27 years in total flying insect biomass in protected areas. *PLOS ONE* **12**, e0185809 (2017) [https://doi.org/ 10.1371/journal.pone.0185809](https://doi.org/10.1371/journal.pone.0185809).
30. Møller, A. P. Parallel declines in abundance of insects and insectivorous birds in Denmark over 22 years. *Ecol. Evol.* **9**, 6581–6587 (2019) [https://doi.org/ 10.1002/ece3.5236](https://doi.org/10.1002/ece3.5236).
31. Van Klink, R. *et al.* Meta-analysis reveals declines in terrestrial but increases in freshwater insect abundances. *Science* **368**, 417–420 (2020) [https://doi.org/ 10.1126/science.aax9931](https://doi.org/10.1126/science.aax9931).
32. FAO. *FAOstat Database*. faostat.fao.org (2023).
33. Garratt, M. P. D. *et al.* Avoiding a bad apple: Insect pollination enhances fruit quality and economic value. *Agric. Ecosyst. Environ.* **184**, 34–40 (2014) [https://doi.org/ 10.1016/j.agee.2013.10.032](https://doi.org/10.1016/j.agee.2013.10.032).
34. Hanley, N., Breeze, T. D., Ellis, C. & Goulson, D. Measuring the economic value of pollination services: Principles, evidence and knowledge gaps. *Ecosyst. Serv.* **14**, 124–132 (2015) [https://doi.org/ 10.1016/j.ecoser.2014.09.013](https://doi.org/10.1016/j.ecoser.2014.09.013).

35. Feuerbacher, A., Herbold, T. & Krumbe, F. The Economic Value of Pollination Services for Seed Production: A Blind Spot Deserving Attention. *Environ. Resour. Econ.* (2024) doi:10.1007/s10640-024-00840-7 [https://doi.org/ 10.1007/s10640-024-00840-7](https://doi.org/10.1007/s10640-024-00840-7).
36. Krumbe, F., Melder, S. & Feuerbacher, A. The Role of Pollination Services in Seed Production: A review. *ecoevorxiv*, (2023) <https://doi.org/https://doi.org/10.32942/X2R61W>.
37. FAO. *State of Food Security Nutrition - Map*. <https://www.fao.org/interactive/state-of-food-security-nutrition/2-1-1/en/> (2024).
38. Breeze, T. D., Gallai, N., Garibaldi, L. A. & Li, X. S. Economic Measures of Pollination Services: Shortcomings and Future Directions. *Trends Ecol. Evol.* **31**, 927–939 (2016) [https://doi.org/ 10.1016/j.tree.2016.09.002](https://doi.org/10.1016/j.tree.2016.09.002).

Dear reviewers,

We sincerely thank you for providing us with many helpful comments and suggestions that have helped us to substantially improve this paper in this second round of review. We greatly appreciate your time and dedication. Below, you will find our detailed response letter. In the revised paper you will also find the main edits marked in red. Minor language and copy-editing changes were not highlighted in the revised paper.

We hope that we have adequately addressed the comments and look forward to your feedback.

Best wishes
The authors

No	Comment	Response
Reviewer 2		
	Dear authors, thank you for addressing all comments. I know we were asking for rather a lot, but I think you could considerably improve the paper. You made the effort of considering a more novel data source, do a sensitivity analysis and you simulate a scenario which as you say has not been simulated like that so far. Therefore, I think your paper has a message worth publishing. Minor comments you should consider before publishing:	We want to thank you for your careful reading and valuable comments. They have helped us greatly to improve the paper, and we appreciate your time and thoughts!
2/1	The model's time line is of 14 years, i.e. starting a simulation in 2016 and simulating a wild pollination collapse in 2030. When mentioning that a pollination collapse cannot be excluded, do you mean in 2030, i.e. in 5 years time or in 14 years of simulation?	Thank you for pointing this out! Yes, we mean by 2030 – because the model's base year is 2017. This corresponds to 13 years (not counting the starting year). We have revised the text to correct for this and to make it clearer overall. See also section 4.1 at L 489 ff.
2/2	Line 116: when using the word "strong yield gap" it would be good to make reference to the table that provides the yield gaps by crop.	Thank you, this has been done. See L127ff.
2/3	Line 175: Clarification of food insecurity has been made, but link to FAO (reference 35) is not available).	The link leads the reader to an interactive map "FAO Hunger map" https://www.fao.org/interactive/state-of-food-security-nutrition/2-1-1/en/ as found on the website: https://www.fao.org/publications/fao-flagship-publications/the-state-of-food-security-and-nutrition-in-the-world/en We have double checked the link and it works on our side. The problem could be caused by different browsers, so we now also cite the FAO report "The State of Food Security and Nutrition in the World"

		https://www.fao.org/interactive/state-of-food-security-nutrition/2-1-1/en
2/4	Line 325: "... regions facing the greatest economic impacts from a pollinator collapse may be less supportive of biodiversity policies, potentially due to concerns about immediate economic costs." Another reason could be the effectiveness of the mentioned biodiversity policies.	Thank you for this thoughtful suggestion. We agree that the effectiveness of existing biodiversity policies could indeed be another important factor influencing voting behavior. However, in our analysis, we focus on the relationship between projected economic vulnerability (as indicated by consumer surplus losses) and political resistance to biodiversity policies. This correlation emerged as an incidental finding and, as noted, warrants further investigation. We acknowledge that multiple causal pathways may underlie this pattern—including perceptions of policy effectiveness, distributional concerns, or broader political economy dynamics—and have now clarified this in the text, see L337ff.
2/5	Discussion or Results could explain why Eastern European countries are more severely affected – i.e. ultimately, this goes back to crop mix, doesn't it?	Yes, this is correct. Countries' relative dependency on the agricultural sector and the crop mix is a major determinant explaining the results. We have added this to the revised paper at L212.

Reviewer 4	
4/1	The research question is really important and timely, in the sense that the data allowing to distinguish the specific impact of wild pollinators have only recently, to my knowledge, acquired a sufficient level of precision and robustness to allow undertaking such an analysis (before reading this article and the cited references, I would have even doubted that this was the case). The hypothesis that wild and managed pollinators "contribute equally to crop pollination" (line 356) may appear as a somewhat clumsy use of the article by Reilly et al (2024) whose use of the results is, as we learn in the "Methods" section (p. 20), ultimately more nuanced. I am not an ecologist, but after a quick read of this article, I note that its authors remain cautious about the robustness of their results and this "equality" estimated on a global scale obviously experiences spatial variations. On the other hand, the hypothesis (line 360-62) that it would be possible to compensate for the disappearance of wild pollinators by developing honeybee colonies does not seem to me to be consistent with the literature on the subject (see : Garibaldi, L. A., Steffan-Dewenter, I., Winfree, R., Aizen, M. A., Bommarco, R., Cunningham, S. A., ... & Klein, A. M. (2013). Wild pollinators enhance fruit set of crops regardless of honeybee abundance. Science, 339(6127), 1608-1611. Requier, F., Pérez-Méndez, N., Andersson, G. K., Blareau, E., Merle, I., & Garibaldi, L. A. (2023). Bee and non-bee pollinator importance for local food security. Trends in ecology & evolution, 38(2), 196-205. Uwingabire, Z., & Gallai, N. (2024). Impacts of degraded pollination ecosystem services on global food security and nutrition. Ecological Economics, 217, 108068. First, we would like to thank you for your thorough review and the many suggestions and comments made. The revised paper has clearly benefitted from these comments, and we hope that we have addressed all points adequately We agree that "equal contribution" is not reflecting the full complexity of the region and crop-specific dependency on wild and managed pollinators, and the substitutability between both. To acknowledge the crop-specific and spatial variation, we have rewritten this statement to (L370): "While the relative contribution of wild and managed pollinators to crop pollination varies across crops (see Table 1) and regions, global evidence suggests that, on average, both groups contribute approximately equally to overall pollination services¹⁹" The following "back-on-the-envelope" calculation of the wild pollinator replacement cost was done following an earlier reviewer's comment. However, we agree that the underlying assumption of substitutability is very strong and simplifying, which requires highlighting. We have made this now clearer in the text by writing (L373 ff.) "Under the strong and simplifying assumption of perfect substitutability between managed and wild pollinators, the loss could theoretically be mitigated by nearly doubling Europe's 25 million honeybee colonies from 2017, which would cost between €2.1 billion and €3.6 billion annually, or approximately €49.7 to €83.4 per hectare for pollination-dependent crops (see Supplementary Methods). While such an approach is consistent with valuation techniques such as beehive rental or replacement cost models⁵¹⁻⁵⁴, it excludes the costs of scaling up other managed pollinators like bumblebees and neglects the likely rise in marginal costs of delivering pollination services. Moreover, the assumption of equal contribution between wild and managed pollinators, although supported by global averages¹⁹, must be treated with caution due to significant spatial and crop-specific variability. In many cases, particularly for crops highly dependent on specific wild pollinator species, substitution is either limited or infeasible^{18,19,55,56}. Therefore, this scenario cannot serve as a realistic estimate of welfare costs and is presented here for illustrative purposes only." We would also note, that there is also literature that shows that for some crops and regions, managed pollinators can substitute wild pollinators, see for instance: https://doi.org/10.1016/j.agee.2024.109135 https://academic.oup.com/jee/article/104/1/107/2199563 https://doi.org/10.1016/j.baae.2021.08.013 https://doi.org/10.1080/14620316.2001.11511320

		We added the calculation following the request of reviewer 1, who is not involved in the review process anymore. While we see some benefit in this calculation, we clearly see its limitations. Therefore, we are open to completely removing it from the manuscript, which leave to the editor to decide.
4/2	A formal point is that it was uncomfortable to read pages of results and discussions on a subject and with an approach, which unavoidably makes the results dependent on the method used to obtain them, before reading anything on this method. This point will probably be useless since this organization is, I believe, a request from the journal, however, it would be useful to give a few words on it at the beginning of the paper. The CAPRI model is now fairly well-known and it seems well suited to modeling the impacts of a pollinator collapse; but it was not obvious to me that it had the capacity to integrate appropriately the impact of fine differences between wild and managed pollinators, including the many substitution that will emerge in the behaviors of the final consumers.	Thank you for this helpful observation. We agree that it can be challenging to interpret results that are model-dependent before having been introduced to the underlying methodology. The structure of the manuscript follows the journal's formatting guidelines, which puts the methods section after the results and discussion. However, we have added a brief explanation of the model's relevance and capabilities early in the manuscript to orient the reader before detailed results are presented. Specifically, we now note that the CAPRI model is well suited to simulate a pollinator collapse scenario, as it allows for behavioral adjustments on both the supply and demand sides—capturing, for instance, shifts by producers and consumers toward crops not dependent on pollination services. We now write in L61ff: “The CAPRI model is well suited to simulate such a scenario, as it allows for adjustments on both the supply and demand sides—including shifts by producers and consumers toward crops that do not rely on pollination services. Its detailed crop-level resolution and regional disaggregation enable the model to capture heterogeneity in pollinator dependence and economic impacts. In addition, CAPRI incorporates trade linkages and price feedback mechanisms, allowing to assess broader implications for food markets, food security, and land use beyond Europe.”
4/3	Along the same lines, the statement (p. 5) that production declines linked to the collapse of pollinators do not translate into a decrease in producers' income by giving several figures, without explaining how they were obtained, is disturbing. Agricultural products often benefit from the "King-Davenant law", but these effects can be erased by the existence of substitute products or in an economy open to imports.	Thank you for this insightful comment. We agree that the link between production losses and producer income gains requires clarification. In response, we have added a sentence immediately following the relevant statement on page 5 (L101) to explain the economic mechanisms behind this outcome. It is important to note, that the results on production and prices already include any substitution and trade effects. Our addition on page 5 (L101) reads the following: “In Europe, the increase in producer prices for crop products by 8.4% (5.8 – 11.2%) overcompensates farmers for the fall in crop yields (Fig. 1), which overall results in producer surplus gains. This result reflects the inelastic nature of agricultural supply

		and demand, which is embedded in the model's parameters, and accounts for substitution effects on both the supply and demand sides as well as trade responses. These mechanisms jointly explain why prices increase more than proportionally to yield losses for certain crops, resulting in income gains for some producers despite falling production (see also the discussion of the King-Davenant-Law below)." Please note that the CAPRI model is not a "net-trade" model but an Armington-type trade model with bilateral trade flows. Under the Armington assumption, domestic and imported goods are treated as imperfect substitutes, with the degree of substitutability governed by an elasticity of substitution (see also the sensitivity analysis). The "strength" of the trade linkage is additionally influenced by the share of total demand met by imports: the higher the import share, the greater the effective substitutability for any given elasticity. We hope this addresses your concern and clarifies the model behavior more transparently for readers.
--	--	--

	Specific comments	
4/4	P.12 lines 230-231. The sentences "... there is a noticeable overlap between the average voting behavior of parliament members of EU member states on two major biodiversity-related proposals (...) This voting behavior has a statistically significant negative correlation with the projected impact of a wild pollinator collapse on consumer welfare..." either say too much or not enough. If you want to learn something from this correlation, I think it's necessary to be more explicit about what it means to you. The observation is, moreover, interesting...	We appreciate your concern with this statement. In our revised manuscript, we clarify that we don't interpret the reported correlation as causal, as the modeled welfare impacts were not known to policymakers at the time of voting. Our intention is rather to highlight a noteworthy structural pattern: member states whose representatives opposed key biodiversity-related legislation also tend to be those most adversely affected in consumer welfare terms under a hypothetical collapse of wild pollination services. This correlation may reflect underlying economic structures or sectoral dependencies—such as the relative importance of pollination-dependent agriculture—rather than any conscious anticipation of such outcomes. We have added this to this section at L248:ff "While the negative correlations do not imply causality, it points to a potential misalignment between ecological vulnerability and political support for biodiversity protection, possibly driven by underlying structural or economic characteristics. Further analysis would be required to disentangle these factors." We would also like to acknowledge that other reviewers advised caution in interpreting this finding. Accordingly, we have revised the relevant passage to explicitly present this result as an exploratory observation, and have avoided attributing any direct behavioural or strategic implications to it at this stage. Please also note that we discuss this observation quite extensively starting in L333 "The significant correlation between consumer surplus losses.". There in L341ff, we also highlight that:

		“This study did not originally set out to examine any political-economic aspects, and it remains limited in scope without investigating causal relationships. Future research could investigate potential confounding factors, such as countries' agricultural dependency as a share of GDP, to better understand why some regions may exhibit higher resistance despite clear economic risks. Additionally, understanding how changes in both consumer and producer surplus interact with policy preferences could offer valuable insights into the socio-economic dynamics of environmental policy support. For instance, while consumer losses correlate negatively with policy support, the positive correlation between producer surplus gains and voting behavior may reflect different economic incentives (Supplementary Figure 4).”
245	P.14 lines 264-265. The statement that a welfare reduction of 3.5% is equivalent to 0.2% GDP is somewhat surprising and could be more explained (we could expect not so different digits).	Thank you, this has been corrected.
4/6	P.17 lines 362-364. The assumption of a “perfect substitutability between managed and wild pollinators” is not supported by the literature (See the references above)	Please see our response and clarifications to comment 4/1 above.
4/7	P.18. The assumption of a complete wild pollinators collapse is unlikely but is not unacceptable as a research postulate to analyze the consequences. On the other hand, the hypothesis of pollination with drones remains, to my knowledge, speculation on an indefinite time horizon, contested by numerous specialists: Potts, S. G., Neumann, P., Vaissière, B., & Vereecken, N. J. (2018). Robotic bees for crop pollination: Why drones cannot replace biodiversity. Science of the total environment, 642, 665-667. Bongomin, O., Gilibrays Ocen, G., Oyondi Nganyi, E., Musinguzi, A., & Omara, T. (2020). Exponential disruptive technologies and the required skills of industry 4.0. Journal of Engineering, 2020(1), 4280156.	We agree that – at least under today’s circumstances – the use of drones seems highly unlikely, and Potts et al (2018) – whom we have already cited - investigate this issue with much more care and level of detail than we could. However, over a 13 year time horizon combined with changes in food prices because of a decline in pollinators, we should at least acknowledge this possibility. Below is the text at L417 in which we have inserted “if ever technically and economic feasible”: “While our model considers adaptation through land-use and trade adjustments, a further limitation is that it does not include technical substitutions, such as hand pollination⁶⁶ or robotic pollinators⁶⁷. However, there is still insufficient data about the cost of these mitigation strategies and particularly robotic pollinators, if ever technically and economic feasible, could cause further ecological and moral risks⁶⁸.” Please note we had already cited the following reference: Potts, S. G., Neumann, P., Vaissière, B. & Vereecken, N. J. Robotic bees for crop pollination: Why drones cannot replace biodiversity. Sci. Total Environ. 642, 665–667 (2018) https://doi.org/10.1016/j.scitotenv.2018.06.114. See reference number 69.
4/8	P.19 lines 427-428. Again, the statement “... with EU members states opposing pollinators-friendly policies carrying disproportionately high burden of impacts” is not precisely explain and remains somewhat disturbing.	We agree that this sentence lacks explanation and context. In response, we have revised the sentence to clarify this as an empirical observation rather than a normative or causal claim. Given that this paragraph concludes the manuscript, we have also adjusted the tone to be more reflective and policy-oriented.

		The revised concluding remark paragraph reads now (L446): Protecting pollinators remains high on the agenda of many governments, particularly within the European Union. At the same time, resistance to biodiversity conservation measures persists. Our results reveal a pronounced regional disparity in the projected economic impacts of a wild pollinator collapse, with several EU member states facing disproportionately high consumer welfare losses. Some of these same countries have shown lower levels of parliamentary support for recent EU biodiversity-related legislation. While this correlation is not causal, it suggests a potential misalignment between ecological vulnerability and political positioning. A key implication for EU institutions may be to design biodiversity and pollinator-friendly policies in a way that better aligns ecological risk exposure with the perceived benefits of such policies, thereby helping to build broader political coalitions for biodiversity protection.
4/9	P. 21 lines 486-487. Again, the acceptability of the technical feasibility of the substitution of wild pollinators by managed pollinators is not explained (and not widely accepted by the literature).	We have revised this text about the substitutability at this section of the manuscript and now mainly address it in the discussion section reflecting the concerns that in most cases there is only limited substitutability if at all. See also our response to your first comment (4/1).
4/10	Line 497. It was a surprise for me to learn there is cocoa nut production in the EU (in overseas territories, I assume)	Cocoa was used as an example, but we agree that it does not fit the European context and therefore it has been removed.